# Privacy-Preserving Personalized Federated Prompt Learning for Multimodal Large Language Models

**Linh Tran**[1]   **Wei Sun**[2]   **Stacy Patterson**[1]   **Ana Milanova**[1]

[1]Rensselaer Polytechnic Institute   [2] IBM Research

## Abstract

Multimodal Large Language Models (LLMs) are pivotal in revolutionizing customer support and operations by integrating multiple modalities such as text, images, and audio. Federated Prompt Learning (FPL) is a recently proposed approach that combines pre-trained multimodal LLMs such as vision-language models with federated learning to create personalized, privacy-preserving AI systems. However, balancing the competing goals of personalization, generalization, and privacy remains a significant challenge. Over-personalization can lead to overfitting, reducing generalizability, while stringent privacy measures, such as differential privacy, can hinder both personalization and generalization. In this paper, we propose a Differentially Private Federated Prompt Learning (DP-FPL) approach to tackle this challenge by leveraging a low-rank factorization scheme to capture generalization while maintaining a residual term that preserves expressiveness for personalization. To ensure privacy, we introduce a novel method where we apply local differential privacy to the two low-rank components of the local prompt, and global differential privacy to the global prompt. Our approach mitigates the impact of privacy noise on the model performance while balancing the tradeoff between personalization and generalization. Extensive experiments demonstrate the effectiveness of our approach over other benchmarks.

## 1 Introduction

In recent years, there has been rapid advancement in multimodal large language models (LLMs) that integrate multiple modality information, including text, images, audio, and video, to enhance the comprehension and generation capabilities. Vision-Language Models (VLMs) such as CLIP (Radford et al., 2021) are a variant of multimodal LLMs that learn transferable image and text representations, making them highly effective in applications such as image captioning and visual search. One proposed setting for deploying VLMs is a Federated Learning framework that allows multiple organizations or clients to collaboratively train a global model without directly sharing their local training data. However, fine-tuning pre-trained VLMs in a FL system is time-consuming and resource-intensive given the massive number of parameters each VLM has. This gives rise to Federated Prompt Learning (FPL) which only fine-tunes the soft prompt embedding while freezing the rest of the VLM model parameters in the FL system (Guo et al., 2023b; Cui et al., 2024). In FPL, each client fine-tunes their customized prompt using their local data and shares the prompt with a central server for generalization purposes. The clients can distribute their fine-tuned prompts as prompt providers to public users who wish to perform downstream inference tasks, also known as the prompt as a service (PaaS) paradigm (Wu et al., 2024; Yao et al., 2024; Huang et al., 2023).

One significant challenge in such distributed systems is the presence of data heterogeneity, i.e., organizations often have non-identical and non-independent (non-IID) data distributions, which can vary widely due to factors such as demographics, usage patterns, or device capabilities. To address this, personalized FL has emerged to tailor models to the unique data characteristics of each client rather than solely improving a global model. Personalized FL focuses on learning customized models for each client, reflecting the heterogeneity of their data (He et al., 2020; Dinh et al., 2020). In the context of personalized FPL, the goal is for each client to learn and utilize personalized prompts that

better align with their specific data and application needs. Nevertheless, over-personalization can lead to local data overfitting, preventing the model to generalize well on non-training data. Clients in an FPL framework may opt to distribute their customized prompts post training to public users for downstream tasks. However, these users may have different types of inputs that the client's prompt is not well generalized to, resulting in suboptimal performance. Consequently, it is crucial to achieve a nuanced balance between personalization and generalization in a heterogeneous FPL system.

In addition to balancing the tradeoff between personalization and generalization, privacy poses another critical concern in FPL, especially in sensitive domains such as finance, law, and healthcare. In the PaaS framework, the distributed trained prompts are shown to be susceptible to Membership Inference Attack (MIA), potentially exposing details about individual clients' training data (Wu et al., 2024). To address this issue, one may consider Differential Privacy (DP) (Dwork et al., 2014) which ensures an adversary cannot reliably detect the presence or absence of a data sample based on the output information. However, balancing personalization and privacy under data heterogeneity is a challenging task. The non-IID nature of the data allows clients to better learn their personalized prompt, but it amplifies the performance degradation caused by DP due to the high data sensitivity, impairing both personalization and generalization capabilities. Thus, the key question we aim to address is: *How can we effectively balance personalization, generalization, and privacy in a data heterogeneous FPL system?*

To tackle the above question, we proposed a Differentially Private Federated Prompt Learning (DP-FPL) approach that leverages low-rank factorization and DP as part of the prompt learning process. In our framework, each client simultaneously learns a global prompt and a local prompt. The global prompt is shared in a FL manner for generalized knowledge transfer, while the local prompt is retained at each client site for personalization. Our contributions are threefold.

- We propose a privacy-preserving personalized federated prompt learning approach with Differential Privacy for multimodal LLMs. We factorize the local prompt into two lower rank components with an additional residual term. The factorized low-rank components allow the model to capture broader patterns that are beneficial across different data distributions, aiding the generalization capability of each client. The residual term is crucial for retaining the expressiveness lost during the factorization process, thereby preserving the client-specific learning and improving personalization.
- We preserve privacy by utilizing both Global Differential Privacy (GDP) and Local Differential Privacy (LDP). Unlike conventional methods that apply noise uniformly to the entire prompt, we judiciously apply LDP to the two low-rank components of the local prompt, and GDP to the global prompt. Our privacy mechanism mitigates the effect of DP noise on model performance while preserving the privacy guarantee post training.
- We conduct extensive experiments on widely adopted datasets to evaluate our proposed method against other benchmarks. The experimental results demonstrate superior performance of our proposed method in balancing personalization and generalization while mitigating the model degradation caused by DP noise.

## 2 RELATED WORK

**Personalized Federated Learning.** There are several existing approaches that aim to learn personalized models for clients in FL settings, including clustering (Ghosh et al., 2020; Berlo et al., 2020; Shahid et al., 2021), regularization (Shoham et al., 2019; Dinh et al., 2020; Li et al., 2020) and knowledge distillation (Li & Wang, 2019; He et al., 2020; Fang & Ye, 2022). Personalized FL is most commonly approached as a multi-task learning problem that simultaneously learns two models for each client: a global model for generalized knowledge and a local model for personalized data. Existing methods accomplish this by decoupling the model parameters or layers into global and local learning components (Arivazhagan et al., 2019; Deng et al., 2020; Zhang et al., 2020; Collins et al., 2021; Jeong & Hwang, 2022; Zhang et al., 2023). In the existing literature on personalized FL, private multi-task learning approaches aim to protect training data by retaining personalized parameters and sharing differentially private generalized parameters. Examples include Jain et al. (2021), Hu et al. (2021), Bietti et al. (2022), Yang et al. (2023b) and Xu et al. (2024). However, these methods are designed for full model training and cannot be directly applied to prompt tuning due to the difference in the parameter space. Sun et al. (2024) incorporates Low-Rank Adaptation with DP

in a standard FL setting, but they do not consider personalization and prompt learning, making their method not applicable to our setting.

**Federated Prompt Learning.** Recent advances in personalized FPL have garnered significant attention (Guo et al., 2023a;b; Li et al., 2023; Yang et al., 2023a; Sun et al., 2023; Deng et al., 2024; Li et al., 2024; Cui et al., 2024). See Table 1 for comparisons. With the exception of Zhao et al. (2023) which introduces a privacy-preserving FPL method that leverages DP to protect the underlying private data, none of the prior literature considers the privacy issue. Many of these works require modification to the backbone model, which is not relevant to our approach as we want to protect the personalized prompt, not the model. Zhao et al. (2023) does not account for the crucial aspects of personalization. Similar to our work, Cui et al. (2024) also factorizes the local prompt into two learnable low-rank components for balancing personalization and generalization. We instead have the learnable full-rank local prompt, and only keep the low-rank terms non-permanent for generalization with an additional residual to retain the expressiveness for personalization.

Table 1: Recent Federated Prompt Learning algorithms

| FPL Algorithm | Consider personalization | No model modification | Adopt low-rank factorization | Provide privacy guarantee |
|---|---|---|---|---|
| pFedPrompt (Guo et al., 2023a) | ✓ | ✗ | ✗ | ✗ |
| PromptFL(Guo et al., 2023b) | ✓ | ✓ | ✗ | ✗ |
| pFedPT (Li et al., 2023) | ✓ | ✗ | ✗ | ✗ |
| pFedPG (Yang et al., 2023a) | ✓ | ✗ | ✗ | ✗ |
| Fedperfix (Sun et al., 2023) | ✓ | ✗ | ✗ | ✗ |
| SGPT (Deng et al., 2024) | ✓ | ✗ | ✗ | ✗ |
| FedOTP (Li et al., 2024) | ✓ | ✓ | ✗ | ✗ |
| FedPGP (Cui et al., 2024) | ✓ | ✓ | ✓ | ✗ |
| Fedprompt (Zhao et al., 2023) | ✗ | ✓ | ✗ | ✓ |
| DP-FPL (ours) | ✓ | ✓ | ✓ | ✓ |

## 3 PROPOSED METHOD

We introduce our proposed method, Differentially Private Federated Prompt Learning (DP-FPL), shown in Figure 1. Our approach leverages low-rank factorization with an additional residual term to balance personalization and generalization in a differentially private FPL system.

### 3.1 PRELIMINARIES ON PERSONALIZED FEDERATED PROMPT LEARNING

Our system follows a standard FPL setting that consists of a set of $N$ clients and a central server. Let the global dataset be $D = [D_1, D_2, \ldots, D_N]$, each client $i$ holds a local subset $D_i$ of $n_i$ samples. Each client local model involves a frozen pre-trained VLM such as a CLIP model and a prompt learner, and their goal is to learn the representation between the visual and prompt information to improve multimodal classification tasks. Specifically, the frozen CLIP model involves a text encoder $f(\cdot)$ and an image encoder $g(\cdot)$ that respectively transform the prompt and an image $x$ into text and image features. The prompt learner trains a soft prompt $p_i$ for client $i$ that is optimized to align with the visual features. Using $\cos[\cdot, \cdot]$ to denote the cosine similarity used by CLIP model, the classification prediction probability for each client $i$ is computed as:

$$\mathbf{p}(\hat{y} = j|x) = \frac{\exp(\cos[f(p_i, c_j), g(x)]/\tau)}{\sum_{k=1}^{C} \exp(\cos[f(p_i, c_k), g(x)]/\tau)}, \tag{1}$$

where $\hat{y}$ denotes the predicted label, $c_j$ denotes label $j$ out of $C$ number of classes, and $\tau$ denotes the temperature parameter of CLIP. The client personalized prompt $p_i$ is optimized with cross-entropy loss:

$$\mathcal{L} = -\frac{1}{|D_i|} \sum_{(x,y) \in D_i} \sum_{j=1}^{C} y \log \mathbf{p}(\hat{y} = j|x). \tag{2}$$

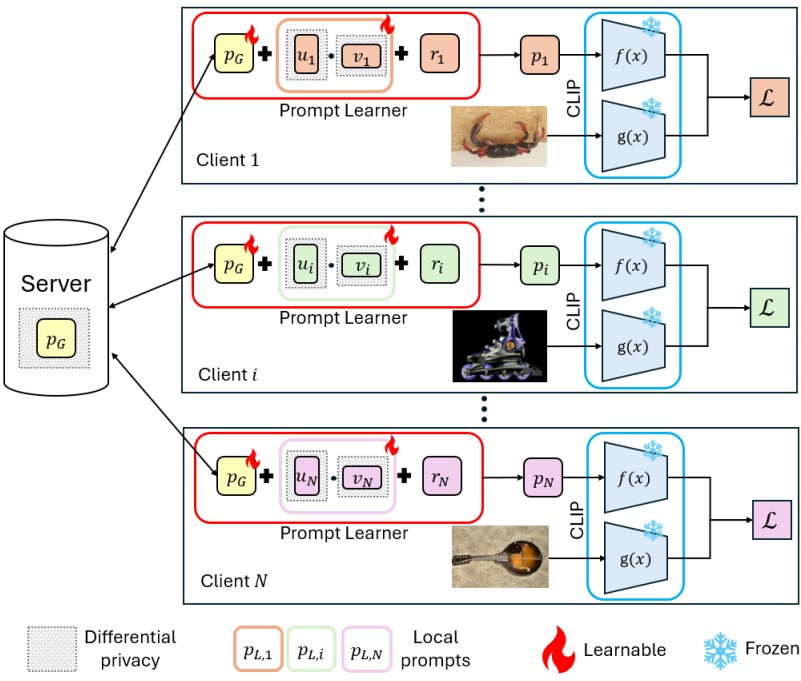

Figure 1: Architecture of DP-FPL with frozen CLIP models. Each client $i$ trains global prompt $p_{G,i}$ and local prompt $p_{L,i}$. The local prompt is factorized at each training iteration as $p_{L,i} = u_i v_i + r_i$.

In a data heterogeneity setting, each client's local dataset $D_i$ is drawn from a distinct data distributions. The difference among clients' data can lead to the drift problem, where the local model converges toward local solutions optimal for their specific data but fails to align with the global model objective. To address this challenge, various personalized FPL solutions such as clustering, local fine-tuning and knowledge distillation have been proposed. Our work focuses on the multi-task learning approach that aims to learn two models for each client: one for generalization and one for personalization. In particular, we separate each client's prompt $p_i$ into a global prompt $p_{G,i}$ and the local prompt $p_{L,i}$. The global prompt $p_{G,i}$ is shared and aggregated to improve the global learning, while the local prompt $p_{L,i}$ is retained at the client level for personalized learning.

A simple example of the personalized prompt is illustrated in Figure 1, in which each client has a collection of images captured from different angles, reflecting the heterogeneity in their local data. Consequently, client 1 might use the prompt "an upside-down photo of [class]", while client $i$ could have the prompt "a upright photo of [class]", and client $N$ might use "a rotated photo of [class]". In this instance, the template "a photo of [class]" is the generalized global prompt shared across clients, and the terms "upside-down", "upright" and "rotated" represent the personalized characteristics of each client's prompt. These variations in prompts allow the client model to better adapt to the specific characteristics of their data, improving personalization.

The training process of FPL over $T$ iterations is structured as follow. For each global training round $t$, each client $i$ initializes the global prompt $p_{G,i}^t \leftarrow p_G^{t-1}$ and the local prompt $p_{L,i}^t \leftarrow p_{L,i}^{t-1}$. Client $i$ then trains their personalized prompt $p_i$ using their local private data and obtains the cross-entropy loss $\mathcal{L}$. At the end of the local training round, client $i$ updates their local prompt $p_{L,i}^t \leftarrow p_{L,i}^t - \eta_{L,i} \nabla_{L,i} \mathcal{L}$ and sends the gradient w.r.t the global prompt $\nabla_{G,i} \mathcal{L}$ to the server for aggregation. The server computes the average gradient $\nabla_G \leftarrow \frac{1}{N} \sum_{i=1}^{N} \nabla_{G,i} \mathcal{L}$ and updates the new global prompt to be $p_G^t \leftarrow p_G^{t-1} - \eta_G \nabla_G$. The learning objective function of FPL system is:

$$\min_{p_G} \sum_{i=1}^{N} \frac{n_i}{\sum_{j=1}^{N} n_j} \mathcal{L}_{D_i}, \tag{3}$$

where $\mathcal{L}_{D_i}$ is the loss computed on dataset $D_i$ of client $i$.

## 3.2 Balancing Personalization and Generalization

Prior research has demonstrated that fine tuning pre-trained LLMs with lower dimension reparameterization promotes generalization capability across various tasks (Aghajanyan et al., 2020; Cui et al., 2024). Additionally, lower intrinsic dimension improves the model utility under the effect of DP noise (Yu et al., 2021; Xu et al., 2024). Therefore, we utilize low-rank factorization as part of our FPL framework to balance the personalization and generalization learning under the influence of DP noise. However, low-rank training such as Low-Rank Adaptation has difficulty matching the performance of full-rank training in many difficult tasks (Liu et al., 2024; Biderman et al., 2024; Ivison et al., 2023; Zhuo et al., 2024). This is because low-rank factorization methods restrict the parameter space, removing some of the information of the full-rank space (Konečný, 2016). Recent work Cui et al. (2024) also used the low-rank approximation in non-private FPL systems, however, they factorize the local prompt only once at the beginning and train iteratively with the low-rank components. This approach can reduce the overall expressive power over the training phase. As we will show in Section 4, the loss of expressiveness can amplify the adverse effects of the DP noise, further diminishing the model's performance.

To overcome these issues of low-rank training, we perform the factorization process in every training round rather than only at the beginning like Cui et al. (2024), and we incorporate a residual term to compensate for the lost expressiveness. We analyze in detail the benefit of the residual term in Section 3.3. The overall personalized prompt of client $i$ can then be expressed as $p_i = p_{G,i} + u_i v_i + r_i$, where $u_i$ and $v_i$ are the factorized low-rank components and $r_i$ is the additional residual term.

We utilize a version of the Reparametrized Gradient Perturbation (RGP) method introduced in Yu et al. (2021) as our low-rank factorization scheme. We perform the factorization process for each client local prompt $p_{L,i}$ in every training iteration to get the temporary low-rank prompt components $u_i$ and $v_i$. We also compute a residual term $r_i$ which is the remainder of the factorization process, i.e. $r_i = p_{L,i} - u_i v_i$. As shown by Yu et al. (2021), each client can reconstruct the gradient of $p_{L,i}$ using the gradient of the low-rank terms in back propagation as follows:

$$\nabla_{L,i}\mathcal{L} = (\nabla_u \mathcal{L})v_i + u_i(\nabla_v \mathcal{L}) - u_i u_i^T (\nabla_u \mathcal{L})v_i \tag{4}$$

We note that the residual $r_i$ is used as part of the forward process, but it is not involved in the full-rank local prompt recomputation.

## 3.3 Preserving Privacy

In the personalized FPL framework, each client as a prompt provider may distribute the trained customized prompt to public users for downstream inference tasks. These fully trained prompts, if not properly protected with privacy mechanism, are vulnerable to MIA which aims to infer if an image sample was used for training or not (Wu et al., 2024). Moreover, the shared global prompt may leak information about the private data during the training process as it contains the gradient information of the loss function.

We assume that clients and server are honest. We consider adversary to be potential user with access to a trained customized prompt obtained from a FPL client. The adversary also has access to the publicly available pre-trained CLIP model for downstream inference purposes. The goal of the adversary is to infer whether an image data was part of the target FPL client's training data utilizing the MIA. In this setting, the target client's trained prompt $p_i$ is shared with a potential adversarial user and is susceptible to privacy breach, so it is necessary to protect $p_i$ with the DP mechanism.

One may straightforwardly apply privacy noise to the client's trained $p_i$ before distributing it to public users. However, this requires a large DP noise to effectively prevent MIA (Wu et al., 2024). On the other hand, injecting noise gradually over the training phase allows for more control over the influence of noise on the model, leading to better utility (Abadi et al., 2016). In our setting, we perform gradient updates on the global prompt $p_{G,i}$ and local prompt $p_{L,i}$ of each client, so we add DP noise to the gradients w.r.t these two terms in each training step.

As part of the FPL procedure, the global prompt of each client $p_{G,i}$ is aggregated and averaged by the server, while the client's local prompt $p_{L,i}$ stays locally. The final trained prompt $p_i$ is composed of the synchronized global prompt $p_G$ and the local prompt $p_{L,i}$. To provide privacy guarantee, we use Global Differential Privacy (GDP) for the global prompt and Local Differential Privacy (LDP)

for the local prompt. This is an unconventional way of introducing DP noise to selective parts of the prompt, unlike a vanilla method that directly adds noise to the whole prompt before publishing it. We define these two DP notions as follow.

**Definition 3.1. Global Differential Privacy (GDP)** A randomized mechanism $\mathcal{M} : \mathcal{D} \to \mathcal{R}$ with domain $\mathcal{D}$ and range $\mathcal{R}$ satisfies $(\epsilon, \delta)$-GDP if for any two adjacent datasets $D, D' \in \mathcal{D}$ (i.e., datasets that differ in exactly one sample) and for any subset of outputs $\mathcal{S} \in \mathcal{R}$ it holds that

$$\Pr[\mathcal{M}(D) \in \mathcal{S}] \leq e^{\epsilon} \Pr[\mathcal{M}(D') \in \mathcal{S}] + \delta$$

**Definition 3.2. Local Differential Privacy (LDP)** A randomized mechanism $\mathcal{M} : \mathcal{D} \to \mathcal{R}$ with domain $\mathcal{D}$ and range $\mathcal{R}$ satisfies $(\epsilon, \delta)$-LDP if for any two adjacent samples $x, x' \in D$ where $D \in \mathcal{D}$ and for any subset of outputs $\mathcal{S} \in \mathcal{R}$ it holds that

$$\Pr[\mathcal{M}(x) \in \mathcal{S}] \leq e^{\epsilon} \Pr[\mathcal{M}(x') \in \mathcal{S}] + \delta$$

We apply GDP to the global prompt $p_{G,i}$ at the server because the impact of the GDP noise on the model utility is much smaller compared to LDP (Arachchige et al., 2019). To provide privacy guarantee for the local prompt $p_{L,i}$ which is not shared with the server for aggregation, each client needs to obfuscate $p_{L,i}$ locally with LDP noise. However, directly applying LDP noise to the full-rank local prompt $p_{L,i}$ can heavily impair the model performance (Yu et al., 2021; Xu et al., 2024). In addition, the local prompt information is predominantly captured within the low-rank components as a result of the factorization process. Therefore, we inject LDP noise only to the low-rank component $u_i$ and $v_i$ to mitigate the negative effect of privacy noise, while still effectively protecting the local prompt during the recomputation process. We note that we do not add noise to the residual component $r_i$ because it is not used for the local prompt recomputation.

We achieve GDP and LDP using the Gaussian noise mechanism. Given a function $f : \mathcal{D} \to \mathcal{R}$, the Gaussian noise mechanism $\mathcal{M}$ is defined as

$$\mathcal{M}(d) \triangleq f(d) + \mathcal{N}(0, \sigma^2) \tag{5}$$

where $\mathcal{N}(0, \sigma^2)$ is the normal distribution with mean 0 and variance $\sigma^2$. The standard deviation $\sigma$ is typically chosen based on the function $f$'s sensitivity $S_f$ to satisfy an $(\epsilon, \delta)$-DP guarantee.

The two main building blocks in our approach, i.e., low-rank factorization and DP, naturally introduce error to the training process. We conjecture that this error acts as a regularization term that prevents clients from overfitting to local data, reducing personalization and improving generalization. However, under strictly private conditions (lower rank and higher DP noise), the accumulated error may become too large and potentially destroy the personalization capability. In this case, the added residual term compensates for the regularization-like error and helps improve local learning, balancing personalization and generalization. We demonstrate the benefit of the residual term in the ablation study in Section 4.3, supporting our hypothesis. Further theoretical analysis of the residual term can be a potential future work direction.

## 3.4 ALGORITHM

We are now ready to describe our proposed method in detail, as shown in Algorithm 1.

At the initial stage, the server randomizes a starting global prompt $p_G^0$ and each client $i$ sets up their starting local prompt $p_{L,i}^0$ (line 1). The variances $\sigma_G$ and $\sigma_L$ are chosen to satisfy a certain $(\epsilon, \delta)$-LDP and $(\epsilon, \delta)$-GDP guarantee. The algorithm runs for $T$ iterations. In each iteration $t$, each client updates $p_{G,i}^t$ to be the previously aggregated $p_G^{t-1}$ and $p_{L,i}^t$ to be the previous $p_{L,i}^{t-1}$ (line 4). A minibatch $\mathcal{B}^t$ is sampled from the local dataset $D_i$ for training (line 5).

Each client performs parameter factorization using the power method with rank $k$ (line 6). Lines $7-8$ describe the forward pass where each client runs their local frozen CLIP model to get the text features and image feature, and uses them to calculate the loss $\mathcal{L}$ using Equation 2. Each client then computes and clips the gradient w.r.t the two low-rank prompt components $\nabla_u \mathcal{L}$ and $\nabla_v \mathcal{L}$ with threshold $C_{th}$ and add local DP noise with standard deviation $\sigma_L$ according to 5 (lines $9-10$).

The noisy gradient w.r.t the local prompt $\tilde{\nabla}_{L,i} \mathcal{L}$ can be reconstructed from the noisy gradients $\tilde{\nabla}_u \mathcal{L}$ and $\tilde{\nabla}_v \mathcal{L}$ using equation 4, and then is updated accordingly (lines $11-12$). Each client also computes

---

**Algorithm 1** DP-FPL

---

1: **Initialize:** $p_G^0, p_{L,i}^0$ for $i = 1 \ldots N$, variances $\sigma_L$ and $\sigma_G$
2: **for** $t \leftarrow 1 \ldots T$ **do**
3:     **for** $i \leftarrow 1 \ldots N$ in parallel **do**
4:         Initialize $p_{G,i}^t \leftarrow p_G^{t-1}$ and $p_{L,i}^t \leftarrow p_{L,i}^{t-1}$.
5:         Sample minibatch $\mathcal{B}^t$ from $D_i$.
6:         Compute low-rank components $u_i, v_i, r_i \leftarrow$ **Factorize**$(p_{L,i}^t, k)$.
7:         Obtain text features $f(p_{G,i}^t)$, $f(u_i v_i + r_i)$ and image feature $g(x)$ $(x \in \mathcal{B}^t)$.
8:         Calculate loss $\mathcal{L}$ according to Equation 2 and compute gradients $\nabla_u \mathcal{L}$ and $\nabla_v \mathcal{L}$.
9:         Clip gradients $\nabla_{G,i} \mathcal{L}$, $\nabla_u \mathcal{L}$ and $\nabla_v \mathcal{L}$ with threshold $C_{th}$.
10:        Add local DP noise: $\tilde{\nabla}_u \mathcal{L} \leftarrow \nabla_u \mathcal{L} + \mathcal{N}(0, \sigma_L^2)$ and $\tilde{\nabla}_v \mathcal{L} \leftarrow \nabla_v \mathcal{L} + \mathcal{N}(0, \sigma_L^2)$.
11:        Recompute noisy gradient w.r.t $\tilde{\nabla}_{L,i} \mathcal{L}$ using $\tilde{\nabla}_u \mathcal{L}$ and $\tilde{\nabla}_v \mathcal{L}$ according to equation 4.
12:        Update local prompt $p_{L,i}^t \leftarrow p_{L,i}^t - \eta_{L,i} \tilde{\nabla}_{L,i} \mathcal{L}$.
13:        Send $\nabla_{G,i} \mathcal{L}$ to the server.
14:     **end for**
15:     Server computes average gradient $\nabla_G \leftarrow \frac{1}{N} \sum_{i=1}^N \nabla_{G,i} \mathcal{L}$.
16:     Server adds global DP noise to $\tilde{\nabla}_G \leftarrow \nabla_G + \mathcal{N}(0, \sigma_G^2)$.
17:     Server updates global prompt $p_G^t \leftarrow p_G^{t-1} - \eta_G \tilde{\nabla}_G$.
18: **end for**

---

the gradient w.r.t the global prompt $\nabla_{G,i} \mathcal{L}$ and sends it to the server for aggregation (line 13). Upon receiving the locally computed gradients, the server computes the average gradient and adds global DP noise with standard deviation $\sigma_G$ to perturb the gradient (lines $15 - 16$). The server then updates the new global prompt for the next training round (line 17).

Low-rank factorization via traditional SVD method requires significant runtime. Instead, we use the power method with one iteration, significantly reducing the computational cost. Given the full-rank matrix of size $m \times n$ (assuming $m \leq n$), the computational cost of SVD scales with $\mathcal{O}(m^2 n)$, while the power iteration only scales with $\mathcal{O}(kmn)$ where $k$ is the reduced rank and $k \ll m$.

### 3.5 PRIVACY ANALYSIS

DP has several properties and compositions that make it easier to analyze the privacy budget in repetitive algorithms such as machine learning, where the privacy loss accumulates across multiple training iterations. When DP mechanisms are applied repeatedly to the same dataset, the overall privacy budget accumulates sequentially using the advanced composition theorem (Dwork et al., 2014). Conversely, when DP mechanisms are applied independently to disjoint subsets of a dataset, the overall privacy loss does not accumulate. In this case, the privacy guarantee remains bounded by the maximum privacy loss of any subset using parallel composition (Dwork et al., 2014). The privacy budget in term of LDP and GDP of Algorithm 1 is given by the following theorem.

**Theorem 3.3.** *There exist constants $c_1, c_2$ so that given the number of global rounds $T$, for any $\delta > 0$, DP-FPL satisfies $(\epsilon, \delta)$-LDP and $(\epsilon, \delta)$-GDP if we choose $\sigma_L$ and $\sigma_G$ as following:*

$$\sigma_L \geq c_1 \frac{S_L \sqrt{T \log(1/\delta)}}{\epsilon} \qquad\qquad \sigma_G \geq c_2 \frac{S_G \sqrt{T \log(1/\delta)}}{\epsilon}$$

*where $S_L = \frac{C_{th}}{|\mathcal{B}|}$ and $S_G = \frac{C_{th}}{N|\mathcal{B}|}$ are the local and global sensitivity respectively.*

*Proof* By definition, a single application of the Gaussian noise mechanism satisfies $(\epsilon, \delta)$-DP if we choose $\sigma \geq \frac{S \sqrt{2 \log(1.25/\delta)}}{\epsilon}$ where $S$ is the sensitivity. Under the advanced composition theorem of DP, the Gaussian noise mechanism after $T$ training steps results in an accumulated privacy loss of $(\mathcal{O}(\epsilon \sqrt{T}), \delta)$-DP. Thus, to achieve $(\epsilon, \delta)$-DP, one would need to choose

$$\sigma \geq c \frac{S \sqrt{T \log(1/\delta)}}{\epsilon} \tag{6}$$

for the Gaussian noise mechanism where $c'$ is a constant.

Applying Equation 6, we can choose $\sigma_L \geq c_1 \frac{S_L \sqrt{T_L \log(1/\delta)}}{\epsilon}$ where $c_1$ is a constant to make Algorithm 1 satisfy $(\epsilon, \delta)$-LDP with respect to each client. Since each client $i$ operates the Gaussian noise mechanism independently on disjoint local subset $D_i$ of the global dataset $D$, the release of all clients' noisy mechanism output still satisfies $(\epsilon, \delta)$-LDP by the parallel composition of DP. Similarly, by choosing $\sigma_G \geq c_2 \frac{S_G \sqrt{T \log(1/\delta)}}{\epsilon}$ according to Equation 6, DP-FPL satisfies $(\epsilon, \delta)$-GDP.

We proved in the theorem above that Algorithm 1 satisfies $(\epsilon, \delta)$-LDP and $(\epsilon, \delta)$-GDP when publishing the customized prompt to potential users. Since the GDP noise is added to the aggregated gradient, the distribution of the aggregated gradient to all clients also satisfies $(\epsilon, \delta)$-GDP by the post-processing property of DP. According to Theorem 3.3, we can calculate the standard deviations $\sigma_L$ and $\sigma_G$ to achieve a certain $(\epsilon, \delta)$-LDP and $(\epsilon, \delta)$-GDP. The pair $\epsilon, \delta$ can be chosen to match a desired MIA accuracy rate, preferably lower than random guess (50%) (Thudi et al., 2022).

## 4 EXPERIMENTS

### 4.1 SETUP

**Datasets.** We select four visual classification datasets to investigate the task of balancing personalization, generalization and privacy: Caltech101 (Fei-Fei et al., 2004), OxfordPets (Parkhi et al., 2012), OxfordFlowers (Nilsback & Zisserman, 2008) and Food101 (Bossard et al., 2014). We utilize the pathological data split among 10 clients. Each client model is trained on its local classes, and evaluated on both its local classes for personalization capability and neighbor classes (classes owned by other clients) for generalization capability. We also evaluate the large-scale dataset CIFAR-100 (Krizhevsky et al., 2009) with the Dirichlet data split among 25 and 50 clients. The final test accuracy is obtained by averaging the performance across all clients. Details of our implementation and hyperparameters are provided in the appendix.

Table 2: Mean test accuracy on local classes averaged across 10 clients. The baseline FedPGP and our method DP-FPL have factorization on the local prompt with rank 8.

| Dataset | Noise $\epsilon$ | PromptFL | FedOTP | FedPGP | DP-FPL |
|---|---|---|---|---|---|
| Caltech 101 | None | 94.45±0.30 | **97.06±0.48** | 95.21±0.19 | 96.06±0.18 |
| | 0.4 | 82.53±0.52 | 95.09±0.50 | 95.12±0.46 | **95.74±0.63** |
| | 0.2 | 81.52±0.45 | 87.61±0.58 | 90.98±0.40 | **95.28±0.46** |
| | 0.1 | 80.42±0.65 | 84.98±0.39 | 80.01±0.39 | **92.71±0.38** |
| | 0.05 | 78.61±1.39 | 83.61±0.38 | 77.24±0.38 | **87.64±0.79** |
| | 0.01 | 78.52±1.68 | 78.86±0.38 | 77.23±0.37 | **85.21±0.85** |
| Oxford Pets | None | 76.85±0.96 | **99.63±0.2** | 94.66±0.31 | 96.91±0.76 |
| | 0.4 | 74.36±0.26 | 80.03±0.93 | 86.56±0.69 | **95.13±0.52** |
| | 0.2 | 73.56±0.16 | 65.97±0.89 | 67.11±0.44 | **93.09±0.43** |
| | 0.1 | 72.77±0.16 | 59.54±0.58 | 63.21±0.62 | **85.25±0.18** |
| | 0.05 | 52.39±0.53 | 58.97±1.02 | 57.98±0.97 | **81.26±1.10** |
| | 0.01 | 43.68±0.67 | 54.08±1.04 | 45.49±1.33 | **73.71±0.40** |
| Oxford Flowers | None | 84.04±0.32 | **97.84±1.16** | 79.11±0.45 | 85.75±0.62 |
| | 0.4 | 60.31±1.28 | 79.89±0.80 | 77.13±0.52 | **80.09±1.41** |
| | 0.2 | 40.33±0.83 | 65.96±0.96 | 70.77±0.61 | **76.75±1.05** |
| | 0.1 | 38.25±1.37 | 42.31±0.71 | 52.42±1.58 | **72.11±1.37** |
| | 0.05 | 37.18±0.92 | 38.89±0.66 | 39.52±0.77 | **69.80±1.34** |
| | 0.01 | 36.11±0.60 | 33.98±0.63 | 35.23±0.64 | **51.55±1.07** |
| Food 101 | None | 86.50±0.26 | **86.65±0.23** | 84.40±0.09 | 86.08±0.12 |
| | 0.4 | 78.70±0.39 | 79.45±0.23 | 80.58±1.52 | **81.45±0.21** |
| | 0.2 | 71.84±0.91 | 77.36±0.41 | 77.72±1.50 | **81.25±0.18** |
| | 0.1 | 69.00±0.40 | 70.48±1.39 | 75.18±0.22 | **80.57±0.46** |
| | 0.05 | 68.36±1.23 | 62.98±1.25 | 73.72±1.04 | **78.23±0.43** |
| | 0.01 | 67.47±1.20 | 54.70±1.12 | 71.82±1.19 | **77.45±0.40** |

**Baselines.** To demonstrate the effectiveness of our proposed method in balancing personalization, generalization and privacy, we compare with three baseline cases: (1) PromptFL (Guo et al., 2023b), (2) FedOTP (Li et al., 2024) and (3) FedPGP (Cui et al., 2024). More details of the baselines are listed in the appendix.

**Privacy levels.** We consider different noise levels for LDP and GDP: $\epsilon = \{0.01, 0.05, 0.1, 0.2, 0.4\}$. We pick $\delta = 10^{-5}$ and the clipping threshold $C_{th} = 10$. We provide details of how the noise is added to each baseline in the appendix.

## 4.2 PERFORMANCE RESULTS

**Improving personalization in private setting.** Table 2 shows the average test accuracy over the last 10 epochs on local classes. In the non-private scenario (i.e., Noise = None), FedOTP shows the highest utility, however, as we add more noise, DP-FPL has the highest local classes accuracy. Under strict privacy levels ($\epsilon = 0.01$), we see a noticeable decrease in test accuracy. Nevertheless, DP-FPL still consistently outperforms other baselines across all datasets, showing the robustness of our method even under strictly private conditions. Table 4 shows the overall accuracy utilizing a Dirichlet data split and ResNet50 as the backbone model. As shown in Table 4, DP-FPL demonstrates superior performance compared to baseline methods across all datasets. This shows the effectiveness and applicability of our approach in various complex settings, including different data distributions, models, and number of clients.

**Privacy improves generalization.** Table 3 shows the average test accuracy over the last 10 epochs on neighbor classes. Similar to local classes, DP-FPL exhibits the highest utility in neighbor classes under the presence of DP noise. As expected, the accuracy degrade for higher noise levels, however, we see an improvement in Caltech101 utility when we increase privacy level from $0.4$ to $0.1$. This is because privacy noise act as a form of regularization that prevents overfitting, and hence improving generalization. Nevertheless, when the noise level is large enough ($\epsilon = \{0.01, 0.05\}$), the overall utility is degraded and the neighbor accuracy no longer improves.

**Additional results.** We include additional experiment on the performance of MIA against DP-FPL in the appendix (see Section A.3). The results show that the attack success rate is relatively low when $\epsilon = 0.1$ for all datasets. In addition, $\epsilon = 0.1$ causes less than $10\%$ reduction in the target model accuracy for both local and neighbor classes as shown in Tables 2 and 3.

Table 3: Mean test accuracy on neighbor classes averaged across 10 clients. The baseline FedPGP and our method DP-FPL have factorization on the local prompt with rank 8.

| Dataset | Noise $\epsilon$ | PromptFL | FedOTP | FedPGP | DP-FPL |
|---|---|---|---|---|---|
| Caltech 101 | None | 92.88±0.19 | 74.91±0.30 | **93.44±0.75** | 91.54±0.15 |
| | 0.4 | 82.66±1.22 | 84.26±0.84 | 88.24±0.78 | **88.58±0.38** |
| | 0.2 | 81.93±1.47 | 84.08±0.98 | 86.05±0.79 | **89.72±0.98** |
| | 0.1 | 80.83±0.72 | 78.91±1.02 | 79.31±0.82 | **90.02±1.47** |
| | 0.05 | 79.96±0.40 | 73.37±1.03 | 75.62±0.82 | **82.76±1.12** |
| | 0.01 | 78.89±0.33 | 73.54±1.00 | 75.60±0.82 | **80.60±0.57** |
| Oxford Pets | None | 76.34±0.37 | 65.63±0.14 | **89.71±0.49** | 80.19±0.85 |
| | 0.4 | 74.43±0.87 | 60.95±0.81 | 72.54±0.54 | **81.82±0.42** |
| | 0.2 | 73.74±0.82 | 59.74±0.84 | 61.68±0.46 | **80.67±0.28** |
| | 0.1 | 73.12±0.39 | 58.83±0.93 | 59.79±0.79 | **77.12±0.52** |
| | 0.05 | 52.44±0.89 | 54.08±0.89 | 51.63±1.01 | **74.13±0.48** |
| | 0.01 | 38.27±0.86 | 53.63±0.91 | 40.20±0.42 | **71.89±0.48** |
| Oxford Flowers | None | 69.44±0.61 | 38.29±1.09 | **75.79±0.87** | 69.51±0.45 |
| | 0.4 | 48.03±0.69 | 56.65±0.77 | 65.93±0.86 | **67.67±0.45** |
| | 0.2 | 38.19±0.97 | 55.44±0.90 | 63.49±0.65 | **67.51±0.58** |
| | 0.1 | 37.76±1.23 | 37.53±1.09 | 46.24±1.86 | **66.44±1.51** |
| | 0.05 | 38.78±1.18 | 33.48±1.00 | 35.27±1.28 | **56.75±1.85** |
| | 0.01 | 34.81±1.19 | 31.26±0.96 | 34.72±1.60 | **43.21±1.72** |
| Food 101 | None | 86.19±0.13 | 84.03±0.33 | **86.23±0.06** | 86.08±0.11 |
| | 0.4 | 76.88±0.23 | 80.11±0.47 | 78.94±1.07 | **81.00±0.25** |
| | 0.2 | 70.99±0.83 | 76.44±0.62 | 77.21±0.90 | **80.79±0.22** |
| | 0.1 | 67.80±1.59 | 73.12±1.65 | 76.92±0.83 | **78.14±0.53** |
| | 0.05 | 66.76±1.27 | 71.82±0.79 | 73.61±1.35 | **77.18±0.50** |
| | 0.01 | 61.42±1.49 | 67.08±1.11 | 72.99±1.53 | **76.87±0.62** |

## 4.3 ABLATION STUDY

In this subsection, we investigate the efficacy of the key parameters that directly affect the tradeoff between personalization, generalization, and privacy: residual term, noise level $\epsilon$ and rank value. The results for Caltech101 are presented in Figure 2; we include other dataset results in the appendix.

**Effect of residual term.** We investigate the effectiveness of the residual term by separately testing the model without the residual component in Figure 2. We note that this setting is different from FedPGP because the factorization process is performed every training round instead of at the beginning. In Figure 2, we observe better performance in both local and neighbor classes when

Table 4: Mean test accuracy averaged across all clients under Dirichlet data distribution. The baseline FedPGP and our method DP-FPL have factorization on the local prompt with rank 8.

| Noise | CIFAR-100 with 25 clients | | | | CIFAR-100 with 50 clients | | | |
|---|---|---|---|---|---|---|---|---|
| $\epsilon$ | PromptFL | FedOTP | FedPGP | DP-FPL | PromptFL | FedOTP | FedPGP | DP-FPL |
| None | 71.20±0.18 | 68.13±0.33 | **71.54±0.17** | 69.84±0.19 | **71.30±0.10** | 68.57±0.14 | 70.82±0.12 | 69.48±0.11 |
| 0.4 | 53.69±0.53 | 47.22±0.97 | 58.87±0.35 | **66.23±0.23** | 55.44±0.31 | 58.23±0.42 | 58.43±0.51 | **66.40±0.18** |
| 0.2 | 53.02±0.39 | 47.02±1.13 | 56.75±0.31 | **62.92±0.30** | 54.65±0.29 | 57.09±0.40 | 56.34±0.44 | **64.39±0.29** |
| 0.1 | 50.86±0.68 | 46.95±1.06 | 55.76±0.33 | **59.53±0.25** | 53.17±1.06 | 57.05±0.22 | 53.39±0.48 | **60.49±0.26** |
| 0.05 | 50.20±0.40 | 44.71±1.16 | 53.80±0.30 | **57.97±0.16** | 53.23±0.83 | 55.00±0.37 | 52.38±0.49 | **58.35±0.30** |
| 0.01 | 50.65±0.41 | 44.27±1.17 | 51.75±0.34 | **53.31±0.35** | 51.92±1.56 | 54.53±0.29 | 52.32±0.46 | **56.09±0.56** |

incorporating the residual term. The difference in accuracy is more prominent under strict privacy conditions (rank 1 and $\epsilon = 0.01$), confirming our conjecture about the benefit of the residual described in Section 3.3. Lower rank and higher noise significantly reduce the overall utility due to the large accumulated error, and the residual term plays a crucial role in improving local learning, balancing personalization and generalization.

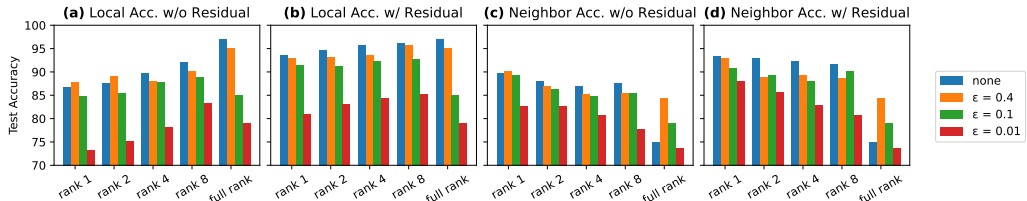

Figure 2: Test accuracy of ablation study on noise level, rank and residual term for Caltech101

**Effect of privacy noise $\epsilon$.** We study how different privacy level affects the model performance on both local and neighbor classes in Figure 2 (b) and (c). As expected, the local classes accuracy gradually decreases when noise level increases. However, we see some unexpected improvement in neighbor utility when we increase DP parameter $\epsilon$ from 0.4 to 0.1. As explained in Section 4.2, certain privacy noise range acts like regularization error that prevents overfitting, resulting in better generalization for higher noise level.

**Effect of factorization rank.** We explore the impact of the factorization rank by comparing different rank values with full-rank setting in Figure 2 (b) and (c). Intuitively, higher rank values lead to better performance, however, this is not always the case. For local classes, rank 8 generally performs better than full-rank under higher noise levels ($\epsilon = \{0.01, 0.1\}$). Lower-rank has fewer entries, hence the amount of noise added is less than full-rank for the same level of privacy guarantee, which leads to better accuracy. For neighbor classes, lower rank value tends to have higher utility because lower rank introduces more regularization error that prevents overfitting and improves generalization.

## 5 CONCLUSION

In this paper, we presented a novel approach to address the critical challenges of personalization, generalization, and privacy in FPL for multimodal LLMs. Our proposed framework leverages low-rank factorization to balance the tradeoffs between these competing objectives. By factorizing the local prompts into low-rank components iteratively while incorporating a residual term, our method effectively preserves both generalization and personalization. Moreover, we introduced a privacy-preserving mechanism that applies both global and local DP to safeguard sensitive client data. Unlike conventional methods, we selectively applied DP noise to the low-rank components, allowing us to maintain privacy without significantly degrading model performance. The critical role of the residual term in mitigating the effects of DP noise was also demonstrated, highlighting its importance for maintaining model expressiveness and personalization.

ACKNOWLEDGMENTS

This work was supported by the IBM through the IBM-Rensselaer Future of Computing Research Collaboration, and by the NSF grant CNS-2232061.

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

# A    EXPERIMENTAL DETAILS

## A.1    DATASETS

**Caltech101** (Fei-Fei et al., 2004) is an image dataset that contains images from 101 object categories (e.g., "helicopter", "elephant" and "chair" etc.). There are about 40 to 800 images for each object category, but most categories have about 50 images. The dataset is available for download on `http://www.vision.caltech.edu/Image_Datasets/Caltech101/101_ObjectCategories.tar.gz`. The dataset contains $6,593$ samples, including $4,128$ training samples and $2,465$ testing samples.

**Oxford Pets** (Parkhi et al., 2012) is an image dataset with 37 object classes that can be downloaded at `https://www.robots.ox.ac.uk/~vgg/data/pets/data/images.tar.gz`. The dataset consists of $6,613$ pet images with roughly 200 images for each class. The dataset is divided into training set of $2,944$ images and test set of $3,669$ images.

**Oxford Flowers** (Nilsback & Zisserman, 2008) is an image classification dataset consisting of 102 flower categories, each class has between 40 and 258 images. The dataset is can be retrieved at `https://www.robots.ox.ac.uk/~vgg/data/flowers/102/102flowers.tgz`. There are $6,556$ total images, including $4,093$ training images and $2,463$ testing images.

**Food101** (Bossard et al., 2014) is a large-scale dataset containing images of 101 different types of food. The dataset is are available for download at `https://data.vision.ee.ethz.ch/cvl/datasets_extra/food-101/`. There are $80,800$ total images, and we split the dataset into train set of size $50,500$ and test set of size $30,300$.

**CIFAR-100** (Krizhevsky et al., 2009) is a another large-scale dataset containing images of 100 different object classes. The dataset is are available for download via `torchvision.datasets.CIFAR10`. The dataset consists of $60,000$ 32x32 images, with $6,000$ images per class. We divide the dataset into training set of $50,000$ images and test set of $10,000$ images.

## A.2    IMPLEMENTATION DETAILS

For the first four datasets Caltech101, OxfordPets, OxfordFlowers and Food101, we use the Vision Transformer ViT-B16 (Dosovitskiy, 2020) as the backbone for the frozen CLIP model for Caltech101, OxfordPets, OxfordFlowers and Food101. For each dataset, we run experiments with $N = 10$ clients for $T = 100$ global training rounds. We use batch size $|\mathcal{B}| = 32$ for training and $|\mathcal{B}| = 100$ for testing. We set the global learning rate $\eta_G = 0.0001$ and local learning rate $\eta_L = 0.0001$ with SGD optimizer.

We adopt the Pathological setting for data heterogeneity among clients as implemented in `https://github.com/KaiyangZhou/CoOp/blob/main/DATASETS.md`. The class labels are splitted randomly among 10 clients without overlapping, and each client owns disjoint set of local classes. For each client, we train the local model on data associated with their assigned local classes. We then evaluate each client's local model on two test sets: local class test set for personalization and neighbor class test set for generalization. The local class test set involve all the test image associated with the client's local class labels. The neighbor class test set is the set of all test images whose labels are owned by other clients.

For CIFAR-100, we adopt ResNet50 (He et al., 2016) as the backbone model and run experiments with $N = 25$ and $N = 50$ clients for $T = 200$ global training rounds. We use batch size $|\mathcal{B}| = 32$ for training and $|\mathcal{B}| = 100$ for testing. We set the global learning rate $\eta_G = 0.0001$ and local learning rate $\eta_L = 0.0001$ with SGD optimizer. We use Dirichlet data distribution to simulate the real-world non-IID setting with parameter $\alpha = 0.3$. The test accuracy is averaged across all clients.

We provide a summary of the experiment set up in Table 5 below.

Table 5: Details of datasets and experimental set up

| Dataset | Data distribution | Model | Number of clients | Number of training rounds | Training batch size | Testing batch size |
|---|---|---|---|---|---|---|
| Caltech101 | Pathological | ViT-B16 | 10 | 100 | 32 | 100 |
| OxfordPets | Pathological | ViT-B16 | 10 | 100 | 32 | 100 |
| OxfordFlowers | Pathological | ViT-B16 | 10 | 100 | 32 | 100 |
| Food101 | Pathological | ViT-B16 | 10 | 100 | 32 | 100 |
| CIFAR-100 | Dirichlet | ResNet50 | 25, 50 | 200 | 32 | 100 |

For the prompt learner, the length of prompt vectors is $b = 16$ with a dimension of $d = 512$, and the token position is "end" with "random" initialization. For the factorization process, we experiment with four different factorization rank $1, 2, 4, 8$. We consider three different DP noises with privacy level from low to high: $\epsilon \in \{0.4, 0.2, 0.1\}$. The clipping threshold is chosen to be $C_{th} = 10$ for both GDP and LDP applications.

For the baseline methods, we consider three settings: PromptFL (Guo et al., 2023b), FedOTP (Li et al., 2024) and FedPGP (Cui et al., 2024) with the main difference lies in the prompt learner structure. We describe each baseline in detail below.

1. **PromptFL:** PromptFL follows the traditional federated learning framework where each client has one single prompt $p_i$ and the aggregated prompt is the average of all clients' $p_i$. In this setting, the privacy noise is added directly to each client's $p_i$ before sharing with the server for aggregation.

2. **FedOTP:** In FedOTP, each client's customized prompt $p_i$ involves two full-rank global prompt $p_{G,i}$ and local prompt $p_{L,i}$. To incorporate privacy, the GDP noise is added to the averaged $p_G$ by the server and the LDP noise is added to each client's local prompt $p_{L,i}$.

3. **FedPGP:** In FedPGP, each client's customized prompt $p_i$ includes a full-rank global prompt $p_{G,i}$ and two low-rank local components $u_i, v_i$. Each client then train three parameters $p_{G,i}$, $u_i$ and $v_i$ across the training process. In this baseline, the GDP noise is added to the averaged $p_G$ by the server and the LDP noise is added to the two low-rank terms $u_i, v_i$ of each client.

We run our experiment on a computer cluster, each node has a 6x NVIDIA Tesla V100 GPUs with 32 GiB of memory and 512 GiB RAM and 2x IBM Power 9 processors. The final result is the mean accuracy across all clients, averaged over 5 runs with different seeds.

## A.3 MEMBERSHIP INFERENCE ATTACK

In this section, we evaluate the privacy-preserving performance of our method DP-FPL against Membership Inference Attack (MIA). We implement MIA as described in Shokri et al. (2017), where the goal of the attack is to infer the appearance of data samples in one client's training dataset. We

first train a set of 50 shadow models with the same model architecture as the target client's model. The shadow models generate synthetic training data that is used to train the attack model. The attack model is a two-layer MLP and a classification head that predicts if a given sample is part of the target client's training data or not. We run the attack on Caltech101, Oxford Pets and Oxford Flowers with different privacy levels. For each dataset, we perform separate attack on each class and compute the average success rate, i.e. percentage of correct guesses.

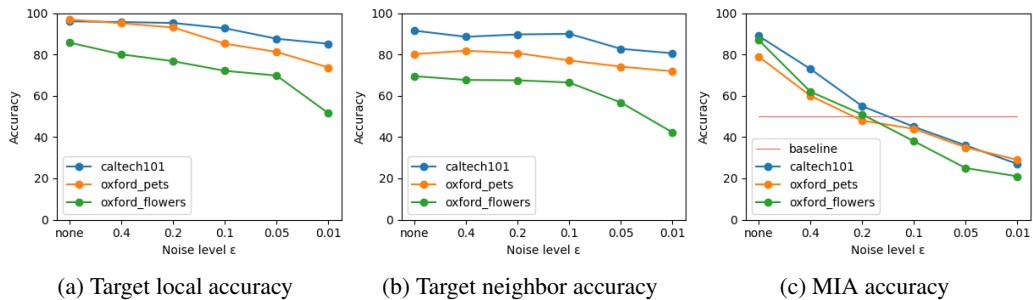

| (a) Target local accuracy | (b) Target neighbor accuracy | (c) MIA accuracy |

Figure 3: Target model performance (a, b) and MIA performance (c) with rank 8. The baseline of MIA accuracy is set to 50% (random guessing).

Figure 3 demonstrates the performance of the target model (local and neighbor classes) and the MIA. We set the MIA baseline accuracy to be 50%, representing the expected success rate of random guessing. Looking at Figure 3c, the MIA accuracy is low ($< 50\%$) when $\epsilon = 0.2$ for Oxford Pets and $\epsilon = 0.1$ for Caltech101 and Oxford Flowers. In addition, $\epsilon = 0.1$ causes minimal loss in the target model accuracy for both local and neighbor classes (Figures 3a and 3b). Therefore, one can balance the utility-privacy tradeoff by setting $\epsilon = 0.1$ for any dataset. This shows that our approach effectively protects the training data from MIA while still maintaining good model performance.

## A.4 ADDITIONAL EXPERIMENTAL RESULTS

In this section, we include additional results from the experiments introduced in Section 4 to further investigate how the DP parameter $\epsilon$, factorization rank and the residual term affect the tradeoff between personalization, generalization and privacy guarantee. Figures 4, 5 and 6 continue the ablation study on the effect of the key parameters that directly affect the tradeoff: residual term, noise level and rank. We summarize the results for each dataset below.

Figure 4 shows the test accuracy with different parameter settings for Oxford Pets. In general, there is an increase in both local and neighbor classes when incorporating the residual term, highlighting the benefit of the residual in model utility in different datasets. The difference in accuracy is more consistent across different noise and rank values compared to Figure 2. In addition, there is minimal growth in accuracy when we increase the rank value. Therefore, it is beneficial to set any rank value for this particular dataset.

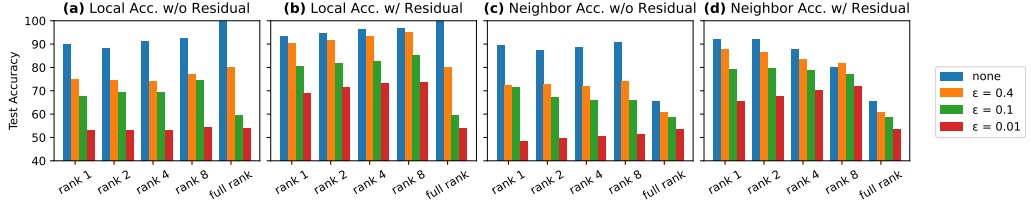

Figure 4: Test accuracy of ablation study on noise level, rank and residual term for Oxford Pets

Figure 5 shows the test accuracy with different parameter settings for Oxford Flowers. The addition of the residual term still improves the overall utility, though the benefit is modest for this dataset. Similar to Oxford Pets, the rank value does not significantly affect the accuracy for both local and neighbor classes. We also observe more drastic drop in accuracy under strong privacy level ($\epsilon =$

0.01). Overall, it is more difficult to balance the tradeoff for Oxford Flowers. One needs to sacrifice strong data protection and set $\epsilon \geq 0.1$ to achieve good personalization and generalization utility.

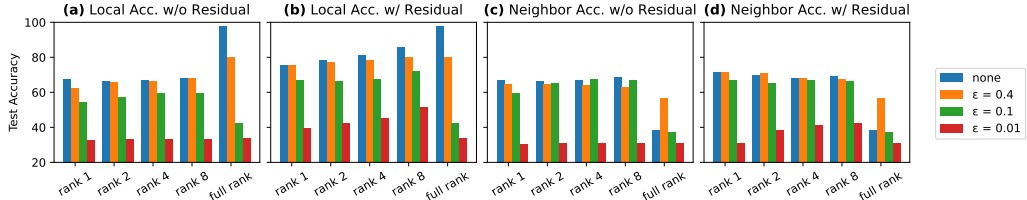

Figure 5: Test accuracy of ablation study on noise level, rank and residual term for Oxford Flowers

Figure 6 shows the test accuracy with different parameter settings for Food101. We see an overall increase in local and neighbor accuracy with the introduction of the residual term, and the difference is more significant under higher noise level. In addition, there is minimal reduction in model utility when we increase privacy noise. This behavior is consistent across all rank values, indicating the robustness of our method under strict privacy constraints.

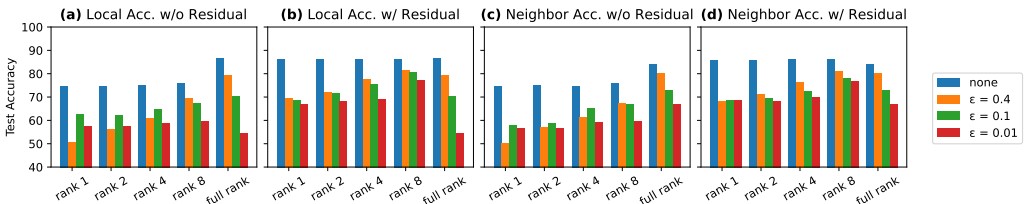

Figure 6: Test accuracy of ablation study on noise level, rank and residual term for Food101

Overall, the affect of the key parameters (noise level, rank and residual term) on the tradeoff between personalization, generalization and privacy varies widely among different datasets. Nonetheless, the results across all datasets show consistent trends, indicating the effectiveness and applicability of our method.

For completeness, we include Tables 6, 7, 8 and 9 which detail the test accuracy with standard deviation of all the ablation experiments demonstrated in Figures 2, 4, 5 and 6.

Table 6: Mean test accuracy of our method averaged across 10 clients under non-private setting (without any DP noise).

| Dataset | Rank | Local classes | | Neighbor classes | |
|---|---|---|---|---|---|
| | | without residual | with residual | without residual | with residual |
| Caltech101 | 1 | 86.64±0.78 | **93.55±0.18** | 89.70±0.81 | **93.40±0.28** |
| | 2 | 87.54±0.83 | **94.51±0.17** | 87.95±0.42 | **92.79±0.72** |
| | 4 | 89.57±0.47 | **95.69±0.57** | 86.86±0.72 | **92.32±0.71** |
| | 8 | 92.12±0.47 | **96.06±0.39** | 87.58±0.17 | **91.54±0.73** |
| Oxford Pets | 1 | 89.78±0.71 | **93.46±0.63** | 89.54±0.71 | **92.17±0.27** |
| | 2 | 88.24±0.73 | **94.49±0.91** | 87.37±0.38 | **91.96±0.71** |
| | 4 | 91.28±0.64 | **96.36±0.28** | 88.70±0.83 | **87.61±0.53** |
| | 8 | 92.42±0.81 | **96.91±0.58** | 90.71±1.09 | **80.19±0.64** |
| Oxford Flowers | 1 | 67.41±0.19 | **75.45±0.20** | 67.22±0.63 | **71.49±0.71** |
| | 2 | 66.18±0.89 | **78.56±0.90** | 66.16±0.10 | **69.76±1.11** |
| | 4 | 67.04±0.61 | **81.53±0.49** | 67.02±0.39 | **68.33±0.32** |
| | 8 | 68.35±0.61 | **85.75±0.91** | 68.75±0.47 | **69.51±0.61** |
| Food101 | 1 | 74.66±1.14 | **86.12±0.38** | 74.58±0.51 | **85.97±0.72** |
| | 2 | 74.91±0.38 | **86.06±0.27** | 74.96±0.49 | **85.89±0.52** |
| | 4 | 74.95±0.91 | **86.18±0.63** | 74.84±0.99 | **86.16±1.06** |
| | 8 | 76.95±1.02 | **86.08±0.95** | 76.02±0.77 | **86.08±0.69** |

Table 7: Mean test accuracy of DP-FPL averaged across 10 clients. The DP noise is set to $\epsilon = 0.4$.

| Dataset | Rank | Local classes | | Neighbor classes | |
|---|---|---|---|---|---|
| | | without residual | with residual | without residual | with residual |
| Caltech101 | 1 | 87.66±0.79 | **92.93±1.18** | 90.19±1.11 | **92.94±0.68** |
| | 2 | 88.98±1.02 | **93.03±0.90** | 86.99±1.34 | **88.82±0.79** |
| | 4 | 88.01±1.21 | **93.58±0.87** | 85.20±1.03 | **89.15±0.90** |
| | 8 | 90.13±1.19 | **95.74±0.63** | 85.46±1.08 | **88.58±0.38** |
| Oxford Pets | 1 | 74.87±0.69 | **90.43±0.79** | 72.41±0.49 | **87.83±0.95** |
| | 2 | 74.66±0.88 | **91.67±0.62** | 72.69±0.33 | **86.26±0.42** |
| | 4 | 74.24±0.74 | **93.22±0.40** | 71.83±0.39 | **83.53±0.60** |
| | 8 | 77.22±0.82 | **95.13±0.52** | 73.86±0.42 | **81.82±0.42** |
| Oxford Flowers | 1 | 62.48±0.83 | **75.45±0.55** | 64.41±0.58 | **71.49±0.55** |
| | 2 | 65.67±0.96 | **77.39±0.69** | 64.53±0.75 | **70.97±0.58** |
| | 4 | 66.23±0.79 | **78.62±1.04** | 63.97±0.77 | **68.10±0.67** |
| | 8 | 68.03±1.41 | **80.09±0.51** | 63.22±0.88 | **67.67±0.45** |
| Food101 | 1 | 50.55±1.05 | **69.76±0.43** | 50.29±1.14 | **68.11±0.94** |
| | 2 | 56.07±0.85 | **72.05±0.66** | 57.10±1.13 | **71.15±0.65** |
| | 4 | 61.02±1.09 | **77.76±0.98** | 61.25±0.58 | **76.48±0.57** |
| | 8 | 69.53±0.20 | **81.45±1.01** | 67.52±0.40 | **81.00±0.90** |

Table 8: Mean test accuracy of DP-FPL averaged across 10 clients. The DP noise is set to $\epsilon = 0.2$.

| Dataset | Rank | Local classes | | Neighbor classes | |
|---|---|---|---|---|---|
| | | without residual | with residual | without residual | with residual |
| Caltech101 | 1 | 87.79±0.81 | **92.16±0.38** | 90.43±0.91 | **91.61±1.02** |
| | 2 | 88.99±0.63 | **93.38±0.82** | 87.22±0.47 | **90.55±0.49** |
| | 4 | 89.25±0.61 | **94.22±0.54** | 85.31±0.28 | **90.73±0.10** |
| | 8 | 91.75±0.19 | **95.28±0.93** | 86.56±0.71 | **89.72±0.83** |
| Oxford Pets | 1 | 70.06±0.73 | **89.79±0.59** | 73.27±0.51 | **87.69±1.08** |
| | 2 | 72.71±0.42 | **90.12±0.49** | 72.04±0.39 | **86.74±0.18** |
| | 4 | 72.27±0.84 | **91.45±0.65** | 70.16±0.77 | **83.62±0.94** |
| | 8 | 73.73±0.94 | **93.09±0.30** | 71.04±0.55 | **80.67±0.69** |
| Oxford Flowers | 1 | 55.18±0.92 | **72.19±0.17** | 64.53±0.81 | **70.35±0.97** |
| | 2 | 59.52±0.53 | **73.58±0.57** | 61.41±0.31 | **70.50±0.29** |
| | 4 | 63.89±0.83 | **75.18±0.57** | 61.50±0.81 | **68.65±0.46** |
| | 8 | 63.86±0.86 | **76.75±0.97** | 62.65±0.65 | **67.51±0.73** |
| Food101 | 1 | 57.38±0.35 | **69.94±0.82** | 58.11±0.37 | **63.63±0.92** |
| | 2 | 59.22±0.16 | **70.52±0.75** | 58.17±0.57 | **67.64±0.38** |
| | 4 | 65.12±0.52 | **75.11±0.46** | 64.74±0.44 | **70.21±0.80** |
| | 8 | 70.16±0.32 | **81.25±0.51** | 70.21±0.75 | **80.79±0.17** |

Table 9: Mean test accuracy of DP-FPL averaged across 10 clients. The DP noise is set to $\epsilon = 0.1$.

| Dataset | Rank | Local classes | | Neighbor classes | |
|---|---|---|---|---|---|
| | | without residual | with residual | without residual | with residual |
| Caltech101 | 1 | 84.73±0.32 | **91.39±0.58** | 89.33±0.85 | **90.81±0.59** |
| | 2 | 85.32±0.38 | **91.11±0.86** | 86.32±0.28 | **89.18±0.67** |
| | 4 | 87.82±0.81 | **92.26±1.28** | 84.73±0.97 | **87.92±0.80** |
| | 8 | 88.82±0.84 | **92.71±0.61** | 85.32±0.86 | **90.02±0.48** |
| Oxford Pets | 1 | 67.59±0.19 | **80.41±0.38** | 71.47±0.32 | **79.24±0.61** |
| | 2 | 69.22±0.48 | **81.65±1.06** | 67.38±0.27 | **79.65±0.59** |
| | 4 | 69.35±0.89 | **82.73±0.49** | 66.11±0.66 | **78.73±0.87** |
| | 8 | 74.48±0.91 | **85.25±0.30** | 66.10±0.49 | **77.12±0.63** |
| Oxford Flowers | 1 | 54.18±0.65 | **66.78±1.02** | 59.51±0.75 | **66.93±0.49** |
| | 2 | 57.16±0.39 | **66.21±0.29** | 65.35±0.64 | **65.21±0.94** |
| | 4 | 59.71±0.83 | **67.32±0.27** | 67.71±0.41 | **67.26±0.32** |
| | 8 | 59.81±0.97 | **72.11±1.05** | 66.98±0.38 | **66.44±0.17** |
| Food101 | 1 | 62.75±0.37 | **68.65±0.81** | 58.05±0.42 | **68.60±0.28** |
| | 2 | 62.43±0.76 | **71.84±0.95** | 58.86±0.26 | **69.66±0.93** |
| | 4 | 64.66±0.75 | **75.64±0.41** | 65.22±0.28 | **72.61±0.30** |
| | 8 | 67.20±0.48 | **80.57±0.83** | 67.06±0.99 | **78.14±0.37** |

Table 10: Mean test accuracy of DP-FPL averaged across 10 clients. The DP noise is set to $\epsilon = 0.05$.

| Dataset | Rank | Local classes | | Neighbor classes | |
|---|---|---|---|---|---|
| | | without residual | with residual | without residual | with residual |
| Caltech101 | 1 | 78.62±0.24 | **81.59±1.55** | 78.75±0.23 | **79.07±1.59** |
| | 2 | 78.63±0.18 | **85.42±0.93** | 77.67±0.34 | **77.42±0.92** |
| | 4 | 78.57±0.48 | **85.05±1.06** | 77.51±0.48 | **82.28±1.07** |
| | 8 | 78.72±0.49 | **87.64±0.98** | 76.83±0.42 | **82.76±1.69** |
| Oxford Pets | 1 | 63.28±0.57 | **70.60±1.30** | 61.75±0.58 | **72.18±1.32** |
| | 2 | 63.32±0.68 | **73.66±0.94** | 61.76±0.70 | **72.62±1.39** |
| | 4 | 64.26±0.91 | **74.29±1.35** | 61.71±0.91 | **68.73±1.31** |
| | 8 | 65.22±0.89 | **81.26±1.00** | 61.69±0.62 | **74.13±0.99** |
| Oxford Flowers | 1 | 32.98±1.13 | **59.62±0.99** | 29.91±0.61 | **46.80±0.91** |
| | 2 | 32.84±1.80 | **59.74±1.05** | 30.02±1.14 | **49.19±1.47** |
| | 4 | 32.45±1.15 | **65.72±1.17** | 30.40±0.75 | **50.32±1.18** |
| | 8 | 33.09±1.62 | **69.80±0.94** | 30.21±1.28 | **56.75±1.23** |
| Food101 | 1 | 58.72±0.36 | **67.19±0.92** | 56.37±0.25 | **67.09±0.46** |
| | 2 | 57.01±0.41 | **67.71±1.42** | 55.59±0.77 | **67.27±0.57** |
| | 4 | 59.03±0.45 | **68.13±1.06** | 55.45±0.19 | **69.44±0.55** |
| | 8 | 64.88±0.37 | **78.23±1.36** | 63.15±0.55 | **77.18±0.77** |

Table 11: Mean test accuracy of DP-FPL averaged across 10 clients. The DP noise is set to $\epsilon = 0.01$.

| Dataset | Rank | Local classes | | Neighbor classes | |
|---|---|---|---|---|---|
| | | without residual | with residual | without residual | with residual |
| Caltech101 | 1 | 73.19±0.25 | **80.89±1.62** | 82.59±0.22 | **87.98±1.59** |
| | 2 | 75.18±0.50 | **83.08±1.16** | 82.60±0.85 | **85.70±0.79** |
| | 4 | 78.19±0.48 | **84.35±1.07** | 80.61±0.48 | **82.81±1.07** |
| | 8 | 83.18±0.60 | **85.21±2.27** | 77.59±0.57 | **80.60±1.12** |
| Oxford Pets | 1 | 53.05±0.59 | **68.73±1.29** | 48.54±1.15 | **65.35±1.36** |
| | 2 | 53.04±1.37 | **71.62±1.10** | 49.54±1.40 | **67.58±1.10** |
| | 4 | 53.06±0.91 | **73.14±1.33** | 50.52±1.35 | **70.39±1.36** |
| | 8 | 54.29±1.01 | **73.71±1.19** | 51.48±1.48 | **71.89±1.48** |
| Oxford Flowers | 1 | 32.44±1.07 | **39.54±0.97** | 30.40±0.90 | **31.24±1.08** |
| | 2 | 33.11±1.84 | **42.53±1.78** | 30.94±1.53 | **38.32±1.34** |
| | 4 | 33.08±1.13 | **45.27±1.23** | 30.91±1.03 | **41.24±1.16** |
| | 8 | 33.15±1.72 | **51.55±1.21** | 30.95±1.80 | **42.31±1.85** |
| Food101 | 1 | 57.63±0.44 | **67.16±0.88** | 56.57±0.31 | **68.79±0.66** |
| | 2 | 57.40±0.43 | **68.28±1.40** | 56.70±1.02 | **68.48±0.60** |
| | 4 | 58.98±0.55 | **69.13±1.05** | 59.39±0.20 | **69.88±0.77** |
| | 8 | 59.63±0.54 | **77.45±3.65** | 59.58±0.73 | **76.87±0.55** |

