# OpenReview forum: "Privacy-Preserving Personalized Federated Prompt Learning for Multimodal Large Language Models"
_ICLR.cc/2025/Conference — ICLR 2025 Poster_

### Official Review · Reviewer_BtXp · 2024-11-02

**Soundness:** 3
**Presentation:** 3
**Contribution:** 2
**Rating:** 5
**Confidence:** 5

**Summary:**

This paper introduces a Differentially Private Federated Prompt Learning (DP-FPL) approach to achieve a performance balance between personalization and generalization in FL setting. Specifically, compared to previous work on this subject, DP-FPL adds a residual term to the low-rank decomposition that achieves better balance between privacy and effectiveness.

**Strengths:**

1. The paper is in general well-written and easy to follow.
2. The effectiveness of proposed framework is testified by comparing with existing works, and improvements are significant.
3. DP theoretical analysis is performed.

**Weaknesses:**

1. Since the introduction of DP part is quite standard for privacy preservation, the novelty mostly lies in the introduction of the residual term compared to FedPGP.

2. Although the standard theoretical analysis on DP is provided, it didn't address the impact of the introduction of the residual term. The paper will be enhanced significantly if it can provide some theoretical insights on the impact of the residual on the utility-tradeoff, which will support its experimental results.

3. The experimental baselines are a bit limited. Only two methods are compared, while many federated prompt learning methods have been proposed in the past，such as PromptFL,pFedPrompt,FedOTP etc.

**Questions:**

If I understand correctly, FedPGP in Table 2 in the experiment section applies the original FedPGP with DP, therefore is the same as "without residual" in Table 3. Correct? In other words, are the result difference between FedPGP and DP-FPL in Table 2 due to the residual term only ?

---

> ### Author Response · Authors · 2024-11-24
>
> Thank you for your detailed feedback and suggestions. We respond to your comments as follows.
>
> *1. Since the introduction of DP part is quite standard for privacy preservation, the novelty mostly lies in the introduction of the residual term compared to FedPGP.*
> -----
> We thank the reviewer for recognizing the significance of our work. While our approach is similar to FedPGP, our low-rank factorization process is inherently different. In FedPGP, the learnable local prompt is **factorized at the beginning** and is kept as low-rank during the entire training process, which reduces the local learning capability due to the lost of expressiveness, as shown in our experimental results. We tackle this issue by **factorizing every training round**, and also incorporating the residual term to retain the error of the factorization process. Furthermore, we introduce an unconventional way of introducing DP noise using both global and local differential privacy to protect the prompt data, unlike a vanilla method that directly adds noise to the whole prompt before publishing it. Our privacy mechanism mitigates the effect of DP noise on model performance while still maintaining the same level of privacy guarantee. In the revised paper, we have emphasized the key differences to highlight our contribution in **Sections 3.2 and 3.3**.
>
> *2. Although the standard theoretical analysis on DP is provided, it didn't address the impact of the introduction of the residual term. The paper will be enhanced significantly if it can provide some theoretical insights on the impact of the residual on the utility-tradeoff, which will support its experimental results.*
> -----
> We thank the reviewer for the thoughtful suggestion, which we address it as follows. Our approach uses low-rank decomposition and DP, and both of these building blocks introduce error to the training process. This error acts as a regularization term that prevents clients from overfitting to local data, reducing personalization and improving generalization. However, under stricter condition (lower rank and higher DP noise), the accumulated error may become too large and destroy the personalization capability. In this case, the added residual term compensates for the regularization-like error and helps improve local learning, balancing personalization and generalization. We have added the discussion of the residual term in the end of **Section 3.3**. In addition, we further experimented with the impact of the residual via the ablation study in **Section 4.3**, and the results support our hypothesis. We leave further theoretical analysis of the residual term as a potential future work direction.
>
> *3. The experimental baselines are a bit limited. Only two methods are compared, while many federated prompt learning methods have been proposed in the past, such as PromptFL, pFedPrompt, FedOTP etc.*
> -----
> We appreciate the reviewer’s feedback regarding the selection of baseline methods. To address the concern of limited baselines, we have expanded our experimental evaluation to include additional baseline methods, specifically **PromptFL** (Guo et al., 2023b) and **FedOTP** (Li et al., 2024). The results of these extended experiments are detailed in Tables 2 to 4, demonstrating the robustness and comparative performance of our proposed method against a broader range of established techniques. Note: To prevent redundancy and streamline our comparisons, we have removed the baseline, FULL, from the updated paper since FedOTP which is also trained with full-rank global and local prompts is a more recent baseline.
>
> Regarding other related works in federated prompt learning, many require modifications to the backbone model, which is not relevant to our approach as we want to protect the personalized prompt, not the personalized model. Our method does not change the backbone model, as outlined in Table 1. As a result, we only consider FedPGP, PromptFL and FedOTP which are directly applicable baselines to our proposed framework.
>
> *4. If I understand correctly, FedPGP in Table 2 in the experiment section applies the original FedPGP with DP, therefore is the same as "without residual" in Table 3. Correct? In other words, are the result difference between FedPGP and DP-FPL in Table 2 due to the residual term only ?*
> -----
> We want to clarify that FedPGP and our method without residual in Table 3 (Table 7 in the revised paper) are different. As explained in our response to point 1, in FedPGP, the learnable local prompt is kept as low-rank during the entire training process, while our method without residual performs the factorization process every training round and excludes the residual term. We have updated the description to make this clear in **Section 4.3**.

---

> > ### Comment · Reviewer_BtXp · 2024-11-27
> >
> > I have read the revised paper and responses, which have partially addressed my concerns. Based on my overall assessment of this paper,   I prefer to maintain my current rating.

---

> > > ### Author Response · Authors · 2024-11-30
> > >
> > > We thank the reviewer for acknowledging our effort in improving our work. During this rebuttal phase, we have:
> > >
> > > - Differentiated our work from others and highlighted the significance and innovation of our method.
> > > - Provided extensive ablation study and detailed discussion on the benefit of the residual term.
> > > - Added more baselines that are directly applicable to our proposed framework.
> > >
> > > We would like to ask if there are any outstanding issues with our paper that are limiting you from considering our paper as acceptable. We appreciate any further feedback you can provide to help us improve our submission, and thank you for your time.

---

### Official Review · Reviewer_U7Cw · 2024-11-03

**Soundness:** 3
**Presentation:** 3
**Contribution:** 3
**Rating:** 6
**Confidence:** 4

**Summary:**

The paper presents a novel approach, Differentially Private Federated Prompt Learning (DP-FPL), which aims to balance privacy, personalization, and generalization in multimodal large language models (LLMs) within federated learning frameworks. DP-FPL utilizes a low-rank adaptation (LoRA) technique with differential privacy (DP) to enable prompt tuning at both global and local levels, allowing clients to personalize prompts without directly sharing sensitive data. This method integrates both local and global differential privacy mechanisms, selectively adding noise to low-rank components to preserve model performance while ensuring data privacy.

**Strengths:**

- The paper combines local and global differential privacy with federated prompt learning, creating a robust privacy-preserving framework suitable for multimodal LLMs.
- By using low-rank adaptation with a residual component, DP-FPL achieves a balance between generalization for broader applicability and personalization for client-specific data.
- The experimental design demonstrates DP-FPL’s effectiveness across multiple datasets and privacy noise levels, providing evidence of the method’s performance under varying conditions.

**Weaknesses:**

- The evaluation lacks statistical tests to confirm the significance of performance differences, which limits the interpretation of the reported improvements.
- The exploration of privacy-performance trade-offs is insufficient under practical, real-world conditions. While the paper discusses differential privacy noise’s impact, it does not fully assess performance under stricter privacy constraints.
- The simulation of data heterogeneity is limited; the paper relies on randomly splitting class labels, which does not capture the complexity of real-world non-IID data distributions effectively.

**Questions:**

- The model shows a performance trade-off between privacy, personalization, and generalization, yet this is not adequately explored in real-world applications. While the authors apply privacy noise at various levels, the impact on performance for real-world, heterogeneous data remains unclear. An experiment demonstrating this trade-off would strengthen the paper.
- The paper experiments with moderate privacy levels (ϵ=0.1,0.2,0.4), but it does not examine stricter levels (ϵ<0.1ϵ < 0.1ϵ<0.1) that may be necessary for sensitive applications like healthcare or finance. Including results for ϵ=0.05 and ϵ=0.01 would provide a fuller understanding of the model’s robustness under high privacy demands.
- The dataset heterogeneity is simulated by randomly assigning class labels to clients, which does not accurately capture real-world, non-IID data distributions. Employing a Dirichlet distribution to split the dataset would more effectively simulate realistic data diversity, enhancing the model's evaluation under practical federated learning conditions.
- Although the paper notes that DP-FPL degrades gradually as DP noise increases, a clearer breakdown of this effect on both personalization and generalization—specifically for local and neighbor classes—across different datasets would add clarity to the trade-offs involved.
- Privacy noise appears to improve generalization by reducing overfitting; however, Table 2 shows mixed effects across datasets and noise levels. For instance, generalization to neighbor classes improves with increased privacy noise on Caltech101 but is inconsistent on Oxford Flowers. Analyzing this inconsistency could provide insights into the regularization benefits of DP noise.

---

> ### Author Response · Authors · 2024-11-24
>
> We thank the reviewer for the constructive suggestions. We address each point raised in the review below.
>
> *1. The evaluation lacks statistical tests to confirm the significance of performance differences, which limits the interpretation of the reported improvements.*
> -----
> We have added standard errors alongside the mean test accuracies in our result tables to illustrate the variability and reliability of our experimental outcomes. To ensure the robustness of our findings, we conducted multiple runs of each experiment and calculated the average performance metrics.
> Due to time constraints, we reported the average run only for Caltech101 in the revised paper. The standard deviations are minimal, indicating consistent performance across multiple runs. We expect the average results for other datasets are consistent as well. We will update the average run results for all other datasets in the final paper.
>
> *2. The exploration of privacy-performance trade-offs is insufficient under practical, real-world conditions. While the paper discusses differential privacy noise’s impact, it does not fully assess performance under stricter privacy constraints.*
> -----
> Thank you for this valuable suggestion. In response, we have conducted additional experiments to evaluate the performance of our method under stricter privacy constraints by using higher noise levels $\epsilon = \{0.01, 0.05\}$. The key findings from these experiments are as follows: at a stringent privacy level of $\epsilon = 0.01$, certain datasets, such as Oxford Flowers, exhibited a noticeable decrease in test accuracy. Despite the increased noise, our method consistently outperforms existing baselines across most datasets. This demonstrates the robustness and effectiveness of our approach in maintaining superior performance even under stricter privacy constraints. The updated results are detailed in Tables 2 to 4 of the revised paper.
>
> *3. The simulation of data heterogeneity is limited; the paper relies on randomly splitting class labels, which does not capture the complexity of real-world non-IID data distributions effectively.*
> -----
> We appreciate the reviewer’s valuable feedback regarding data heterogeneity. To more accurately represent real-world non-IID data distributions, we conducted additional experiments on the CIFAR-100 dataset using a Dirichlet distribution with parameter $\alpha = 0.3$. This approach better captures the variability and complexity inherent in practical scenarios. The new results, detailed in Table 4 of the revised manuscript, are consistent with our original findings and further validate the effectiveness and robustness of our proposed method under more realistic heterogeneous conditions.

---

> ### Author Response · Authors · 2024-11-24
>
> *4. The model shows a performance trade-off between privacy, personalization, and generalization, yet this is not adequately explored in real-world applications. While the authors apply privacy noise at various levels, the impact on performance for real-world, heterogeneous data remains unclear. An experiment demonstrating this trade-off would strengthen the paper.*
> -----
> As mentioned in our response to point 3, we conducted further experiments on a real-world non-IID data distribution using a Dirichlet data split. We added detailed discussion and analysis about the trade-off between accuracy and privacy for the new experiments in **Section 4.2**.
>
> *5. The paper experiments with moderate privacy levels $(\epsilon=0.1,0.2,0.4)$, but it does not examine stricter levels $(\epsilon<0.1\epsilon<0.1\epsilon<0.1)$ that may be necessary for sensitive applications like healthcare or finance. Including results for $\epsilon=0.05$ and $\epsilon=0.01$ would provide a fuller understanding of the model’s robustness under high privacy demands.*
> -----
> As mentioned in our response to point 2, we have conducted more experiments with stricter noise level $\epsilon = \{ 0.01, 0.05 \}$. The results (Tables 2 to 4) show that our approach is generally more robust than other baselines even under high privacy levels.
>
> *6. The dataset heterogeneity is simulated by randomly assigning class labels to clients, which does not accurately capture real-world, non-IID data distributions. Employing a Dirichlet distribution to split the dataset would more effectively simulate realistic data diversity, enhancing the model's evaluation under practical federated learning conditions.*
> -----
> As mentioned in our responses to points 3 and 4, we have conducted further experiments using Dirichlet data split per your request, and we reported the results in Table 4.
>
> *7. Although the paper notes that DP-FPL degrades gradually as DP noise increases, a clearer breakdown of this effect on both personalization and generalization—specifically for local and neighbor classes—across different datasets would add clarity to the trade-offs involved.*
> -----
> We have added more in depth ablation study on the effect of the noise level as well as the rank and residual term to further study the trade-off between privacy and accuracy (local and neighbor classes). We reported the ablation experiments for Caltech101 dataset in Figure 2 in the main paper, and included the results for other datasets in the appendix. To summarize, higher DP noise generally degrades both local and neighbor classes accuracy for most datasets. However in some special cases, certain noise ranges can improve generalization for specific datasets by preventing overfitting to local classes. This behavior is highly dependent on the data sensitivity of the dataset. In addition, if the noise is too large, the overall utility will degrade with no improvement in generalization capability.
>
> *8. Privacy noise appears to improve generalization by reducing overfitting; however, Table 2 shows mixed effects across datasets and noise levels. For instance, generalization to neighbor classes improves with increased privacy noise on Caltech101 but is inconsistent on Oxford Flowers. Analyzing this inconsistency could provide insights into the regularization benefits of DP noise.*
> -----
> This is an excellent observation, we thank the reviewer for pointing this out. We note that it is expected that the accuracy (for both local and neighbor classes) will degrade as we increase the privacy level. However, there is an atypical behavior for Caltech101 dataset where higher privacy noise improves generalization. We hypothesize that this is because privacy noise act as a form of regularization, and this behavior may be distinct for every dataset due to the difference in data sensitivity. Nevertheless, if the noise level is large enough, the overall utility will degrade and we will not have the benefit of generalization. To further demonstrate our conjecture, we ran additional experiments for Caltech101 with higher noise level $\epsilon = \{ 0.01, 0.05 \}$. The results (Table 3 in revised paper) show that the neighbor accuracy no longer improves under too strict privacy constraints. We have updated the discussion to illustrate this point in **Section 4.2**.

---

> > ### Comment · Reviewer_U7Cw · 2024-11-26
> >
> > Thanks for the authors' responses. The authors have addressed my concerns and I will maintain my rating of acceptance for the paper.

---

> > > ### Author Response · Authors · 2024-11-30
> > >
> > > Thank you for your feedback and engagement with our work. We appreciate your recognition of the extended experiments we have conducted on more complex settings, including strict privacy constraints and real-world non-IID data distribution. The additional empirical results have provided more insight into the effectiveness of our method, especially on the relationship between generalization and privacy. We appreciate your time and feedback which has helped improve the overall quality of our paper.

---

### Official Review · Reviewer_Xnij · 2024-11-04

**Soundness:** 3
**Presentation:** 3
**Contribution:** 2
**Rating:** 6
**Confidence:** 3

**Summary:**

This article focuses on the issues of balancing personalization, generalization, and privacy in Federated Prompt Learning, and proposes a Differentially Private Federated Prompt Learning method to solve them, and finally proves the effectiveness of the proposed method in three datasets.

**Strengths:**

1. It is very important and meaningful to consider privacy in Federated Prompt Learning.
2. The proposed method is simple and easy to deploy.
3. Existing methods are discussed and compared in detail.
3. The privacy guarantee of the proposed method is given.

**Weaknesses:**

1. Although I appreciate the simplicity of the proposed method, compared with existing methods, it seems that only the residuals and differential privacy are added, which are commonly used strategies, making the novelty insufficient.
2. Inadequate experiments. First, the baselines are limited, and it is unclear why a range of baseline methods from related work was not used. Secondly, it is unclear about the privacy-preserving performance of the proposed method, such as its performance in the face of membership inference attacks. Furthermore, no significance tests were performed. Finally, the proposed method has a large number of components, such as global prompt, local prompt, GDP, LDP, etc., all of which require ablation experiments.
3. There are a large number of assertions that have not been confirmed and are all the author's personal conjectures. For example, "simply applying LoRA can result in loss of expressiveness during training, ...", " this approach does not protect the privacy of the prompt data..., Therefore, privacy noise must be incorporated into the gradient updates during each training step for privacy guarantee", "the impact of the GDP noise on the model utility is much smaller compared to LDP".
4. Lack of complexity discussion.
5. Minor: What is $\epsilon$ in Table 4?

**Questions:**

Please refer to Weaknesses.

---

> ### Author Response · Authors · 2024-11-24
>
> Thank you for your thoughtful review and detailed feedback. Below we address each question and concern raised.
>
> *1. Although I appreciate the simplicity of the proposed method, compared with existing methods, it seems that only the residuals and differential privacy are added, which are commonly used strategies, making the novelty insufficient.*
> -----
> We appreciate the reviewer’s recognition of the simplicity of our method. While it is true that our approach incorporates residual connections and differential privacy—techniques that are individually well-established—the novelty of our work emerges from the innovative combination of these strategies to effectively balance personalization, generalization, and privacy.
>
> 1. **Integration of Personalization, Generalization, and Privacy:**
>    - **Differential Privacy (DP):** DP provides robust data protection but can compromise model performance due to the added noise. To mitigate this, we employ matrix factorization to reduce the impact of noise on the model's efficacy. This approach builds on the foundation laid by Yu et al. (2021), who demonstrated the effectiveness of factorization in noise mitigation but did not address the interplay between personalization and generalization.
>    - **Factorization Enhancement:** Unlike FedPGP, our method modifies the factorization process to occur in every training round rather than only at the beginning. This continuous update ensures a dynamic adjustment that better accommodates the evolving model parameters, enhancing both personalization and generalization over time.
>
> 2. **Residual Connections for Improved Local Learning:**
>    - We incorporate a residual term to promote local learning capabilities. This addition specifically addresses the limitation observed in Cui et al. (2024), where factorization improved generalization but potentially reduced local learning performance. By integrating residual connections, our method effectively balances the benefits of factorization with enhanced local adaptation, ensuring that personalization is not sacrificed for generalization.
>
> 3. **Innovative Differential Privacy Mechanism:**
>    - Our approach introduces a hybrid differential privacy mechanism that utilizes both global and local differential privacy. Unlike conventional methods that apply noise uniformly to the entire prompt, our dual approach strategically distributes noise addition. This technique not only preserves the overall privacy guarantees but also minimizes the adverse effects of noise on model performance, achieving a more refined balance between privacy and utility.
>
> In summary, our method distinguishes itself by **combining residual connections and differential privacy in a novel framework** that simultaneously enhances personalization, generalization, and privacy. This integrated approach addresses the shortcomings of existing methods and offers a more balanced and effective solution for protecting prompt data without compromising model performance. We believe these contributions substantively advance the current state of the art.
>
> Da Yu, Huishuai Zhang, Wei Chen, Jian Yin, and Tie-Yan Liu. Large scale private learning via low-rank reparametrization. In *International Conference on Machine Learning*, pp. 12208–12218. PMLR, 2021.
>
> Tianyu Cui, Hongxia Li, Jingya Wang, and Ye Shi. Harmonizing generalization and personalization in federated prompt learning. In *Forty-first International Conference on Machine Learning*, 2024. URL https://openreview.net/forum?id=YYwERRXsJW.

---

> ### Author Response · Authors · 2024-11-24
>
> *2. Inadequate experiments. First, the baselines are limited, and it is unclear why a range of baseline methods from related work was not used.*
> -----
> We appreciate the reviewer’s feedback regarding the selection of baseline methods. Our primary baseline, FedPGP, was chosen because it closely aligns with our methodology through its use of low-rank decomposition, and it is considered state of art for addressing personalization and generalization, making it a highly relevant point of comparison.
>
>
> To address the concern of limited baselines, we have expanded our experimental evaluation to include additional baseline methods, specifically **PromptFL** (Guo et al., 2023b) and **FedOTP** (Li et al., 2024). The results of these extended experiments are detailed in Tables 2 to 4, demonstrating the robustness and comparative performance of our proposed method against a broader range of established techniques. Note: To prevent redundancy and streamline our comparisons, we have removed the baseline, FULL, from the updated paper since FedOTP which is also trained with full-rank global and local prompts is a more recent baseline.
>
> Regarding other related works in federated prompt learning, many require modifications to the backbone model, which is not relevant to our approach as we want to protect the personalized prompt, not the personalized model. Our method does not change the backbone model, as outlined in Table 1. As a result, we only consider FedPGP, PromptFL and FedOTP which are directly applicable baselines to our proposed framework.
>
> Tao Guo, Song Guo, Junxiao Wang, Xueyang Tang, and Wenchao Xu. Promptfl: Let federated participants cooperatively learn prompts instead of models-federated learning in age of foundation model. *IEEE Transactions on Mobile Computing*, 2023b.
>
> Hongxia Li, Wei Huang, Jingya Wang, and Ye Shi. Global and local prompts cooperation via optimal transport for federated learning. *arXiv preprint arXiv:2403.00041*, 2024.
>
> *3. Secondly, it is unclear about the privacy-preserving performance of the proposed method, such as its performance in the face of membership inference attacks.*
> -----
> The reviewer raises a valid question. Previous study has shown that one can derive the bound of the success rate of membership inference attack (MIA) on an $(\epsilon, \delta)$-differentially private training algorithm (Thudi et al., 2022). The MIA accuracy is generally lower than random guess ($50\%$) for appropriately chosen privacy parameters $\epsilon, \delta$. We added this statement in our privacy analysis in **Section 3.5**.
>
> Anvith Thudi, Ilia Shumailov, Franziska Boenisch, and Nicolas Papernot. Bounding membership inference. *arXiv preprint arXiv:2202.12232, 2022*.
>
> *4. Furthermore, no significance tests were performed.*
> -----
> In the revised paper, we have significantly expanded our experimental evaluation to enhance the robustness and comprehensiveness of our findings. Specifically, we conducted a wide range of experiments across multiple small-scale and large-scale datasets under various complex settings, including different data distributions, models, and numbers of clients (**Section 4.2**). To deepen the understanding of our method, we performed more extensive ablation studies that investigate the influence of each key parameter (**Section 4.3**). Additionally, we have now included standard deviations alongside the mean test accuracies in our result tables to provide a clearer picture of performance variability. A thorough description of our experimental setup and ablation studies can be found in the appendix.
>
> *5. Finally, the proposed method has a large number of components, such as global prompt, local prompt, GDP, LDP, etc., all of which require ablation experiments.*
> -----
> We want to clarify that the components global prompt and local prompt are the learnable parameters of the training algorithm, and the notion of GDP and LDP are the privacy guarantees we provide given the adversary model described in **Section 3.3**. These components are part of the training and privacy-preserving objectives, and hence cannot be changed. We studied the key parameters that directly affect the tradeoff between personalization, generalization and privacy: noise level $\epsilon$, rank value and residual term. We identified which components are key parameters and have discussed more in depth the ablation experiment for each component in **Section 4.3**.

---

> ### Author Response · Authors · 2024-11-24
>
> *6. There are a large number of assertions that have not been confirmed and are all the author's personal conjectures. For example, "simply applying LoRA can result in loss of expressiveness during training, ...", " this approach does not protect the privacy of the prompt data..., Therefore, privacy noise must be incorporated into the gradient updates during each training step for privacy guarantee", "the impact of the GDP noise on the model utility is much smaller compared to LDP".*
> -----
> We thank the reviewer for highlighting these concerns. We apologize for the lack of sufficient explanations and evidence supporting our claims in the initial submission. Below, we address each point in detail and have incorporated the necessary revisions into the updated papers:
>
> 1. *"simply applying LoRA can result in loss of expressiveness during training, ..."*
>
>    Prior research has demonstrated that LoRA often struggles to match the performance of full fine-tuning on several challenging tasks (Liu et al., 2024; Biderman et al., 2024; Ivison et al., 2023; Zhuo et al., 2024). This limitation stems from the fact that LoRA, along with other low-rank decomposition methods, constrains the parameter space by discarding some information inherent in the original full-rank space (Konecny, 2016). We have updated this statement in the manuscript to reflect these findings (see **Section 3.2**).
>
> 2. *" this approach does not protect the privacy of the prompt data..., Therefore, privacy noise must be incorporated into the gradient updates during each training step for privacy guarantee"*
>
>    We recognize that our previous assertion was overly definitive. To clarify, adding noise at the final stage of training does offer a degree of data protection. However, introducing noise incrementally throughout the training process provides better control over its impact on the model, thereby enhancing utility (Abadi et al., 2016). Additionally, Wu et al. (2024) indicates that a substantially larger privacy budget (e.g., $\epsilon = 20$) is necessary to effectively safeguard data against membership inference attacks when noise is only added at the final step, which negatively affects model utility. Consequently, incorporating noise during each training step is a more widely adopted and preferable method. We have revised our manuscript to include this clarification and supporting evidence (see **Section 3.3**).
>
> 3. *"the impact of the GDP noise on the model utility is much smaller compared to LDP"*
>
>     In federated learning, GDP noise is applied to the aggregated gradient, while LDP noise is applied more frequently to each client to achieve individual data privacy. Due to this, LDP often requires more noise to provide the same privacy protection as GDP, resulting in more severe degradation to the model accuracy (Arachchige et al., 2019). This statement is updated in **Section 3.3**.
>
> Shih-Yang Liu, Chien-Yi Wang, Hongxu Yin, Pavlo Molchanov, Yu-Chiang Frank Wang, Kwang-Ting Cheng, and Min-Hung Chen. Dora: Weight-decomposed low-rank adaptation. *arXiv preprint arXiv:2402.09353*, 2024.
>
> Dan Biderman, Jacob Portes, Jose Javier Gonzalez Ortiz, Mansheej Paul, Philip Greengard, Connor Jennings, Daniel King, Sam Havens, Vitaliy Chiley, Jonathan Frankle, Cody Blakeney, and John Patrick Cunningham. LoRA learns less and forgets less. *Transactions on Machine Learning Research*, 2024. ISSN 2835-8856. URL https://openreview.net/forum?id=aloEru2qCG. Featured Certification.
>
> Hamish Ivison, Yizhong Wang, Valentina Pyatkin, Nathan Lambert, Matthew Peters, Pradeep Dasigi, Joel Jang, David Wadden, Noah A Smith, Iz Beltagy, et al. Camels in a changing climate: Enhancing lm adaptation with tulu 2. *arXiv preprint arXiv:2311.10702*, 2023.
>
> Terry Yue Zhuo, Armel Zebaze, Nitchakarn Suppattarachai, Leandro von Werra, Harm de Vries, Qian Liu, and Niklas Muennighoff. Astraios: Parameter-efficient instruction tuning code large language models. *arXiv preprint arXiv:2401.00788*, 2024.
>
> Jakub Konecny. Federated learning: Strategies for improving communication efficiency. *arXiv preprint arXiv:1610.05492*, 2016.
>
> Martin Abadi, Andy Chu, Ian Goodfellow, H. Brendan McMahan, Ilya Mironov, Kunal Talwar, and Li Zhang. Deep learning with differential privacy. In *Proceedings of the 2016 ACM SIGSAC Conference on Computer and Communications Security*, CCS ’16, pp. 308–318. Association for Computing Machinery, 2016.
>
> Yixin Wu, Rui Wen, Michael Backes, Pascal Berrang, Mathias Humbert, Yun Shen, and Yang Zhang. Quantifying Privacy Risks of Prompts in Visual Prompt Learning. In *USENIX Security Symposium (USENIX Security)*. USENIX, 2024.
>
> Pathum Chamikara Mahawaga Arachchige, Peter Bertok, Ibrahim Khalil, Dongxi Liu, Seyit Camtepe, and Mohammed Atiquzzaman. Local differential privacy for deep learning. *IEEE Internet of Things Journal*, 7(7):5827–5842, 2019.

---

> ### Author Response · Authors · 2024-11-24
>
> *7. Lack of complexity discussion.*
> -----
> Low-rank decomposition requires heavy computational runtime if we use SVD. However, in our method we use the power method (one iteration) for the decomposition process, which significantly reduces the computational cost. Given the original full-rank matrix of size $m \times n$ (assuming $m \leq n$), the computational cost of SVD scales with $\mathcal{O}(m^2 n)$, while the power iteration only scales with $\mathcal{O}(kmn)$ where $k$ is the reduced rank and $k \ll m$. We have added the complexity discussion in **Section 3.4**.
>
> *8. Minor: What is $\epsilon$ in Table 4?*
> -----
> Table 4 (Table 6 in revised version) shows the performance of our method without any differential privacy noise (non-private setting), so no $\epsilon$ was used in this experiment. We have updated the caption to make this clear.

---

> > ### Comment · Reviewer_Xnij · 2024-11-25
> >
> > Thanks to the authors for the detailed response, most of my questions were well addressed and I increased the score accordingly. However, I am still concerned about the gap between theoretical privacy guarantee and actual privacy performance. The paper cited by the authors is also not peer-reviewed. It would be better if this part of the experiment could be supplemented.

---

> ### Author Response · Authors · 2024-11-30
>
> We thank the reviewer for engaging in the rebuttal and raising their score. To address your concern about the privacy-preserving performance of our method, we have conducted further experiment with Membership Inference Attack (MIA) for three datasets: Caltech101, Oxford Pets and Oxford Flowers. The detailed implementation is described in the appendix of the revised paper along with the experimental results and discussion. We summarize our findings as follows.
>
> Figure 3 (<https://ibb.co/1vh2Q25>) describes the target model accuracy on local classes (a) and neighbor classes (b), and the success rate of the MIA (c) for all three datasets.
> We observe in Figure (c) that the success rate is low (less than the random guessing baseline 50\%) when $\epsilon = 0.1$ for all datasets. In addition, $\epsilon = 0.1$ causes less than 10\% reduction in the target model accuracy for both local and neighbor classes as shown in Figures (a) and (b). This shows that our approach effectively protects the training data from MIA while still maintaining good model performance, balancing the utility-privacy tradeoff.
>
> We thank the reviewer for your useful comments and suggestions. We would appreciate any further feedback you can provide to help us improve our submission, and thank you for your time.

---

> > ### Comment · Reviewer_Xnij · 2024-11-30
> >
> > I thank the authors for the additional experiments and I have no further questions. I encourage their inclusion in the main text. I am willing to raise the score again.

---

> > > ### Author Response · Authors · 2024-11-30
> > >
> > > We thank the reviewer for your prompt response and for raising the score. We are glad that we have addressed all of your questions with the addition of the ablation study, membership inference attack, more baseline comparison and complexity analysis. We appreciate your helpful feedback and suggestions which have helped significantly strengthen our paper, and thank you for your time.

---

### Official Review · Reviewer_BrD2 · 2024-11-04

**Soundness:** 2
**Presentation:** 3
**Contribution:** 2
**Rating:** 6
**Confidence:** 4

**Summary:**

This paper proposes a Differentially Private Federated Prompt Learning (DP-FPL) system for multimodal large language models to address the critical need of balancing the competing goals of personalization, generalization, and privacy-preserving that are involved in serving those models. By decoupling the globally learnable prompt from the personal learnable prompt, decomposing the local prompt with LoRA with an additional residual term, and employing differential privacy techniques in both local prompt learning and global prompt aggregation, DP-FPL achieves accuracy gains while meeting specific privacy budget in a distributed and heterogenous data setting.

**Strengths:**

The motivation of the paper is well stated, and writing is easily understood. The problem the paper tries to address is certainly important and practical in multimodal LLM deployment. The experiment setting is clear, and results support the claim the authors make.

**Weaknesses:**

While the problems the paper tries to address are certainly important, the paper does not provide significant scientific insight nor rigorous analysis that can help solve them in broad situations. Instead, a specific set of existing techniques are shown to be effective in a narrow setting (i.e., small-scale datasets and one model architecture). Though the introduction of the residual term in the local prompt decomposition is empirically demonstrated to be effective in the experimental settings, the paper does not give theoretical justification of why such design is beneficial and how applicable it is when applying to more complex situations (e.g., larger data heterogeneity). The privacy analysis also lacks rigorous proof of why the global gradient aggregation satisfies the DP bound.

**Questions:**

Are there more results on more clients, large datasets, and other models beside ViT-B16? Any theoretical justification of why the residual term preserves the expressiveness of the local prompt?

---

> ### Author Response · Authors · 2024-11-24
>
> Thank you for taking the time to review our paper. Below we respond to the weaknesses and questions raised.
>
> *1. While the problems the paper tries to address are certainly important, the paper does not provide significant scientific insight nor rigorous analysis that can help solve them in broad situations. Instead, a specific set of existing techniques are shown to be effective in a narrow setting (i.e., small-scale datasets and one model architecture).*
> -----
> Thank you for your valuable feedback. To address your concerns regarding the breadth and rigor of our analysis, we have performed several additional experiments and in-depth studies:
>
> 1. **Expanded Large-Scale Dataset Evaluation**:
>     - **Food101**: We implemented our method using the ViT-B16 architecture as the backbone model.
>     - **CIFAR-100**: We employed the ResNet50 architecture to evaluate our approach.
>
>     These large-scale datasets demonstrate that our method maintains its effectiveness beyond small-scale settings, showcasing its applicability across diverse and more complex data scenarios.
>
> 2. **Comprehensive Ablation Studies**:
>     We conducted detailed ablation studies focusing on key parameters that directly affect the tradeoff between personalization, generalization and privacy, including:
>     - **Noise Level** for privacy protection
>     - **Rank** for low-rank factorization
>     - **Residual Term**
>
>     These studies provide deeper insights into the influence of each parameter on our method's performance, highlighting the robustness and adaptability of our approach.
>
> 3. **Consistent and Robust Results**:
>     The new experimental outcomes are consistent with our initial findings, further validating the effectiveness and reliability of our method across different architectures and larger datasets. The ablation experiment also shows the importance of the inclusion of the residual term in balancing personalization, generalization and privacy.
>
> All the new empirical results and detailed analyses are presented in **Sections 4.2 and 4.3** of the revised manuscript. These additions not only enhance the scientific rigor of our work but also demonstrate its applicability in broader and more varied settings.
>
> We believe that these comprehensive evaluations and analyses significantly strengthen our paper, providing the necessary scientific insights and rigorous validation to address the concerns raised.
>
>
> *2. Though the introduction of the residual term in the local prompt decomposition is empirically demonstrated to be effective in the experimental settings, the paper does not give theoretical justification of why such design is beneficial and how applicable it is when applying to more complex situations (e.g., larger data heterogeneity).*
> -----
> Our approach uses low-rank decomposition to balance personalization and generalization and differential privacy (DP) to protect the sensitive prompt. Both of these building blocks introduce error to the training process. We hypothesize that this error acts as a regularization term that prevents clients from overfitting to local data, reducing personalization and improving generalization. However, under strictly private conditions (lower rank and higher DP noise), the accumulated error may become too large and potentially destroy the personalization capability. In this case, the added residual term compensates for the regularization-like error and helps improve local learning of the local prompt, balancing personalization and generalization. We have added the analysis in the end of **Section 3.3**, as well as additional empirical results for more complex settings as described previously in **Sections 4.2 and 4.3**.
>
> *3. The privacy analysis also lacks rigorous proof of why the global gradient aggregation satisfies the DP bound.*
> -----
> We assume that the server and clients are honest, and the adversaries are public users with access to a fully trained customized prompt obtained from a FPL client, as described in the prompt as a service paradigm. Therefore, we care about the privacy guarantee of the published customized prompt. Nevertheless, because we apply global differential privacy (GDP) to the aggregated gradient with the privacy budget analyzed in Theorem 3.3, the distribution of the aggregated gradient to all clients satisfies $(\epsilon, \delta)$-GDP with the same privacy budget by the post-processing property of DP. We added this statement to the privacy analysis in **Section 3.5**.

---

> ### Author Response · Authors · 2024-11-24
>
> *4. Are there more results on more clients, large datasets, and other models beside ViT-B16?*
> -----
> As mentioned in our response to point 1, we conducted additional experiments on two large datasets Food101 and CIFAR-100, using Vit-B16 as the backbone model for Food101 and ResNet50 for CIFAR-100. We have also increased the number of clients to $25$ and $50$ for CIFAR-100. All new experiments show similar and consistent results with the existing empirical setup. The new results are reported in **Section 4.2**.
>
> *5. Any theoretical justification of why the residual term preserves the expressiveness of the local prompt?*
> -----
> We refer the answer to our response to point 2 above. We have added the discussion of the residual term in **Section 3.3**. Our hypothesis is supported by the ablation experiment results in **Section 4.3**. We leave further theoretical analysis of the residual term as a potential future work direction.

---

> > ### Comment · Reviewer_BrD2 · 2024-11-26
> > **Post-rebuttal Comments**
> >
> > Thanks for the authors' feedback. The new results and discussions address most of my concerns. I raise my rate for respect.

---

> > > ### Author Response · Authors · 2024-11-30
> > >
> > > We thank the reviewer for recognizing our effort and raising their score. We are glad that we have addressed your concerns with the new experiments on more complex settings, including larger datasets, different model, more number of clients and extended ablation study. The new empirical results have demonstrated the applicability of our method in more diverse settings.
> > >
> > > We also appreciate your recognition of our extended discussion on the benefit of the residual term and the privacy bound of the global gradient. We thank the reviewer again for your constructive feedback and suggestions that have helped enhance the quality of our paper.

---

### Author Response · Authors · 2024-11-24

We thank the reviewers for your time and effort in reviewing our paper. We appreciate your thoughtful feedback and suggestions, and we carefully address each comment and question in our response. All updates and changes are incorporated and highlighted in blue in our revised paper.

---

### Author Response · Authors · 2024-11-30

We thank the reviewers for their helpful comments and suggestions. We are glad to see that the reviewers appreciate the following aspects of our work:

- Importance of the problem (BrD2, Xnij)
- Strong motivation of the problem (BrD2)
- Innovation of our proposed method (U7Cw)
- Extensive theoretical DP analysis (Xnij, BtXp)
- Strong experimental results supporting our claims (BrD2, U7Cw, BtXp)

We summarize the updates we made on our manuscript to incorporate the reviewers' feedback and suggestions:

- Additional experiments in different complex settings, including larger datasets, different model, different data distribution, more number of clients and stricter privacy constraints (BrD2, Xnij, U7Cw).
- Extended ablation study on the DP noise level, factorization rank value and the residual term to further demonstrate their effect on the tradeoff between personalization, generalization and privacy (BrD2, Xnij, U7Cw).
- A detailed discussion about the benefit of the residual term in our method, which is supported by the extensive ablation study (BrD2, BtXp).
- Addition of baselines that are relevant to our work for comparison. The results show that our method is more effective and robust against privacy constraints (Xnij, BtXp).
- Evaluation of our proposed method against membership inference attack. The results show that our approach is effective in defending the training data with less than 10\% reduction in the target model accuracy, balancing the utility-privacy tradeoff (Xnij).

We note that all of the updates have been reflected and highlighted in blue in the revised paper. We thank the reviewers again for taking the time to review our paper and providing constructive feedback that has helped strengthen our submission.

---

### Meta-Review · Area_Chair_bXPh · 2024-12-19

**Metareview:**

The paper explores Differentially Private Federated Prompt Learning (DP-FPL) for multimodal large language models, examining it from the perspectives of personalization, generalization, and differential privacy. The authors provide a largely straightforward theoretical analysis and present experimental results. However, they do not offer a theoretical rationale for the newly introduced residual term, even though its effectiveness is supported by an ablation study.

**Additional Comments On Reviewer Discussion:**

There were relatively few experimental baselines, a concern partially addressed during the discussion phase. Initially, many terms and assertions were undefined or uncited, which the authors addressed in their rebuttal—for instance, the citation for LDP was only added in the revised manuscript, causing some confusion about the paper’s novelty. The authors also included an ablation study to further demonstrate the effectiveness of the new residual term in the objective.

---

### Decision · Program_Chairs · 2025-01-22

Accept (Poster)